

# On the Witten index of 3d $\mathcal{N} = 2$ unitary SQCD with general CS levels

**Cyril Closset and Osama Khlaif**

School of Mathematics, University of Birmingham, Watson Building,
Edgbaston, Birmingham B15 2TT, United Kingdom

## Abstract

We consider unitary SQCD, a three-dimensional $\mathcal{N} = 2$ supersymmetric Chern-Simons-matter theory consisting of one $U(N_c)_{k,k+lN_c}$ vector multiplet coupled to $n_f$ fundamental and $n_a$ antifundamental chiral multiplets, where $k$ and $l$ parameterise generic CS levels for $U(N_c) = (SU(N_c) \times U(1))/\mathbb{Z}_{N_c}$. We study the moduli space of vacua of this theory with $n_a = 0$, for generic values of the parameters $N_c, k, l, n_f$ and with a non-zero Fayet-Ilopoulos parameter turned on. We uncover a rich pattern of vacua including Higgs, topological and hybrid phases. This allows us to derive a closed-form formula for the flavoured Witten index of unitary SQCD for any $n_f \neq n_a$, generalising previously known results for either $l = 0$ or $n_f = n_a$. Finally, we analyse the vacuum structure of recently proposed infrared-dual gauge theories and we match vacua across the dualities, thus providing intricate new checks of those dualities. Incidentally, we also discuss a seemingly new level/rank duality for pure CS theories with $U(N) \times U(N')$ gauge group.

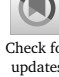

# 1   Introduction

Supersymmetric quantum field theories often admit rich families of degenerate ground states preserving supersymmetry – the vacuum moduli spaces. Given a supersymmetric theory $\mathcal{T}$, many tools exist to characterise its moduli space of vacua, $\mathcal{M}[\mathcal{T}]$. One such tool is the Witten index, $\mathbf{I}_W$ [1]. The index is essentially the Euler characteristic of the moduli space, appropriately defined:

$$\mathbf{I}_W[\mathcal{T}] = \chi\left(\mathcal{M}[\mathcal{T}]\right). \tag{1.1}$$

In this work, we study the moduli space of vacua of the 3d $\mathcal{N} = 2$ supersymmetric unitary SQCD theory with generic CS levels. In particular, we compute its Witten index. This generalises a number of previous results [2–5]. See also *e.g.* [6–17] for related works.

The Witten index is obtained as a trace of the fermion number operator $(-1)^F$ over the Hilbert space of the theory compactified on a torus, and as such it only receives contributions from the zero-energy states [1]. As written here, the index is only well-defined in the absence of non-compact directions in moduli space – so that the 'Euler characteristic' (1.1) makes sense. In this work, we are interested in 3d $\mathcal{N} = 2$ field theories whose vacuum structure depends continuously on a choice of 'real masses'. The Witten index is then well-defined for generic-enough real-mass deformations, and at the same time it is independent of the specific choice of the masses [3]. In the geometric language, the mass deformation corresponds to an 'equivariant' deformation of $\mathcal{M}[\mathcal{T}]$, so that the index (1.1) remains well-defined. Equivalently, we simply turn on fugacities for flavour symmetries when taking the trace over the Hilbert space, hence $\mathbf{I}_W[\mathcal{T}]$ is now called the flavoured Witten index:

$$\mathbf{I}_W[\mathcal{T}] = \mathrm{Tr}\left((-1)^F y^{Q_F}\right), \tag{1.2}$$

with $y$ denoting the flavour fugacities for the flavour charges $Q_F$ that commute with supersymmetry.

Unitary SQCD$[N_c, k, l, n_f, n_a]$ is the 3d $\mathcal{N} = 2$ supersymmetric gauge theory consisting of a $U(N_c)_{k,k+lN_c}$ vector multiplet, where $k$ and $l$ parameterise the CS levels for the $SU(N_c)$ and $U(1)$ factors, coupled to $n_f$ fundamental chiral multiplets and $n_a$ antifundamental chiral multiplets. In a previous work [5], we studied and clarified recently-proposed infrared dualities [18, 19] which generalised well-known Seiberg-like [20] dualities from the $l = 0$

case [21–23] to the general $l \in \mathbb{Z}$ case. In that work, we also computed the Witten index, in principle, as the number of Bethe vacua of the theory compactified on a circle. However, the Bethe vacua computation relied on Gröbner basis algorithms that are computationally intensive, and they could not be used to give a closed-form formula for the index in general.

In this paper, we compute the index for unitary SQCD completely explicitly. Along the way, we uncover an intricate structure of vacua at non-zero values of the Fayet-Iliopoulos (FI) parameter. An explicit expression for the SQCD index was obtained in [5] only for the case $n_f = n_a$. Here we address the case $n_f \neq n_a$ following a two-step strategy. First, we will show that the index satisfies a recursion relation such that, for $n_f > n_a$, the index is given by:

$$\mathbf{I}_W[N_c, k, l, n_f, n_a] = \sum_{j=0}^{n_a} \binom{n_a}{j} \mathbf{I}_W[N_c - j, k, l, n_f - n_a, 0]. \tag{1.3}$$

Next, we will compute the index of SQCD with $n_a = 0$. This theory is very 'geometric' in nature – in fact, for 'small enough' CS levels, the moduli space of vacua with positive FI parameters, $\xi > 0$, is given by the Grassmannian manifold:

$$\mathcal{M}_{\text{Higgs}} = \text{Gr}(N_c, n_f), \tag{1.4}$$

and the index is then, literally, its Euler characteristic. In general, in addition to this Higgs branch, there will be a rich structure of topological and hybrid Higgs-topological vacua, hence the 'moduli space' $\mathcal{M}[\mathcal{T}]$ is really a union of ordinary Higgs branches and of 'non-geometric' vacua – namely, topological quantum field theories (TQFTs) in the guise of 3d $\mathcal{N} = 2$ pure Chern-Simons (CS) theories. Topological vacua of 3d $\mathcal{N} = 2$ CS-matter theories [3] were studied in closely related contexts in *e.g.* [16,17]; 3d $\mathcal{N} = 1$ versions of some of the non-abelian topological and hybrid vacua discussed here also appear on domain walls of 4d $\mathcal{N} = 1$ SQCD [24,25].

To carry out this computation, we will simply study the semi-classical vacuum equations [3]; those are the classical vacuum equations with one-loop corrections to the CS levels and FI parameters. For generic $n_f$ and $n_a$, they take the form:

$$(\sigma_a - m_i) Q_i^a = 0, \qquad (-\sigma_a + \widetilde{m}_j) \widetilde{Q}_a^j = 0,$$
$$\sum_{i=1}^{n_f} Q_a^{\dagger i} Q_i^b - \sum_{j=1}^{n_a} \widetilde{Q}_j^{\dagger b} \widetilde{Q}_a^j = \frac{\delta_a{}^b}{2\pi} F_a(\sigma; \xi, m), \tag{1.5}$$

where $m_i$, $\widetilde{m}_j$ are real masses for the squarks $Q^i$, $\widetilde{Q}^j$, and $F_a$ are specific linear functions. When considering the case $n_a = 0$, we will then mainly focus on the case $m_i = 0$ and $\xi \neq 0$. (The real masses $m_i$ correspond to $SU(n_f)$-equivariant deformations of the Grassmannian (1.4) along its isometries.) This allows us to fully determine the moduli space, except in the so-called marginal case with $|k| = \frac{n_f}{2}$ – in that case, whenever $\xi k l > 0$, certain strongly-coupled vacua are expected to contribute, as we will explain. In any event, we can compute the full Witten index even in those cases.

As we vary the FI parameter from positive to negative values, the vacuum structure changes dramatically but the Witten index is expected to remain the same [3], as indeed we confirm by explicit computation when $|k| \neq \frac{n_f}{2}$. For example, for particular choices of the CS levels $k$ and $l$, we could have a pure geometric Higgs branch (1.4) for $\xi > 0$, while we may obtain a pure TQFT for $\xi < 0$. This observation generalises previous computations in the abelian case [3]. It is also reminiscent of the standard Landau-Ginzburg/Calabi-Yau correspondence for 2d GLSMs that flow to 2d SCFTs, which is also obtained by varying the (exactly marginal, 2d) FI parameter [26]. It would be interesting to explore this analogy further.

Finally, we also consider the vacuum moduli spaces of the gauge theory that are conjectured to be infrared dual to SQCD$[N_c, k, l, n_f, 0]$. We find a perfect matching between vacua in the two dual descriptions,[1] which can be viewed as an intricate cross-check on those dualities. Interestingly, the matching of the TQFT vacua between the two sides is only apparent, in general, after taking into account certain infrared dualities for pure 3d $\mathcal{N} = 2$ CS theories – so-called level/rank dualities –, including some new level/rank dualities that we discuss below. (Those new dualities follow implicitly from the general discussion in [27].)

The analysis presented here allows us to completely characterise the 'geometric window' for which a 3d GLSM to the Grassmannian $Gr(N_c, n_f)$ exists – these are the theories without any topological of hybrid vacua for $\xi > 0$. Those 'Grassmannian theories' have interesting enumerative geometry interpretations [28–32] which we will revisit in future work [33].

This paper is organised as follows. In section 2, we first compute the Witten index of certain unitary pure $\mathcal{N} = 2$ CS theories, and we discuss their level/rank dualities. In section 3, we recall the semi-classical description of 3d $\mathcal{N} = 2$ SQCD$[N_c, k, l, n_f, n_a]$ and we give an overview of the types of vacua that one can obtain in those theories (especially for $n_a = 0$); we also review some well-known results in the abelian case. We then derive the recurrence relation for the Witten index of SQCD$[N_c, k, l, n_f, n_a]$. In section 4, we solve the semi-classical vacuum equations for $U(N_c)_{k, k+lN_c}$ coupled to $n_f$ fundamental chiral multiplets. In section 5, we discuss the matching of vacua implied by various 3d $\mathcal{N} = 2$ infrared dualities. In appendix A, we briefly discuss the non-supersymmetric versions of the level/rank dualities of section 2. Some additional examples of matching across dualities are collected in appendix B. See also the attached MATHEMATICA notebook, wherein all our results are implemented explicitly.

# 2 3d $\mathcal{N} = 2$ pure Chern-Simons theories: Witten index and dualities

In this section, we compute the Witten index of $\mathcal{N} = 2$ supersymmetric pure Chern-Simons theories with unitary gauge groups. We also introduce some level/rank dualities for those theories, including an interesting duality for a $U(N) \times U(N')$ theory which will be useful in later sections.

## 2.1 Witten index of CS theories with unitary gauge group

Consider the 3d $\mathcal{N} = 2$ Chern-Simons theory:

$$U(N_1)_{k_1, k_1+l_1 N_1} \times U(N_2)_{k_2, k_2+l_2 N_2} \times \cdots \times U(N_n)_{k_n, k_n+l_n N_n}, \tag{2.1}$$

where we allow all possible mixed CS levels $k_{ij}$ between different gauge groups, namely:

$$\underbrace{U(N_i)_{k_i, k_i+l_i N_i} \times U(N_j)}_{k_{ij}}{}_{k_j, k_j+l_j N_j}, \tag{2.2}$$

for $i \neq j$. Let us first recall that we have the following decomposition into $SU(N)$ and $U(1)$ factors:

$$U(N)_{k, k+lN} \cong \frac{SU(N)_k \times U(1)_{N(k+lN)}}{\mathbb{Z}_N}. \tag{2.3}$$

The $\mathcal{N} = 2$ $SU(N)_k$ theory has a Witten index [2, 34]:

$$\mathbf{I}_W[SU(N)_k] = \binom{|k| - 1}{N - 1}, \tag{2.4}$$

---

[1] At least for $|k| \neq \frac{n_f}{2}$ and $l + \text{sign}(k) \neq 0$. The remaining special cases are left as challenges for the future.

while for the $U(N)$ theory we find [5]:

$$\mathbf{I}_W\left[U(N)_{k,k+lN}\right] = \frac{|k+lN|}{N}\binom{|k|-1}{N-1}. \tag{2.5}$$

The mixed Chern-Simons levels in (2.2) only involve the $U(1)$ factors. In the normalisation (2.3), this corresponds to:

$$U(1)_{\underbrace{N_i(k_i+l_iN_i)}} \times U(1)_{N_j(k_j+l_jN_i)}. \tag{2.6}$$
$$\underbrace{\phantom{U(1)_{N_i(k_i+l_iN_i)}}}_{N_iN_jk_{ij}}$$

Let us denote by $\boldsymbol{K} \equiv (K_{ij})$ the matrix of CS levels for the abelian sector:

$$K_{ij} = \begin{cases} N_i(k_i+l_iN_i), & \text{if } i=j, \\ N_iN_jk_{ij}, & \text{if } i \neq j. \end{cases} \tag{2.7}$$

**Index for the abelian theory.** Recall that a $U(1)_K$ CS theory has a Witten index

$$\mathbf{I}_W[1,K] = |K|, \tag{2.8}$$

which is of course a special case of (2.5). For any abelian CS theory with gauge group $U(1)^n$ a matrix of CS levels $\boldsymbol{K}$, the Witten index is the number of Bethe vacua, which is given by the number of solutions to the system of equations:

$$q_i(-x_i)^{K_{ii}}\prod_{j\neq i}x_j^{K_{ij}} = 1, \qquad i = 1,\cdots,n. \tag{2.9}$$

Here $q_i$ is the fugacities for the topological symmetries and $x_i$ are the gauge variables, in the conventions of [5,35]. Then, the Witten index is the absolute value of the determinant of the CS level matrix $\boldsymbol{K}$, which we write as:

$$\mathbf{I}_W[\mathbf{1}\,\mathbf{0}\,|\,\boldsymbol{K}] = |\det \boldsymbol{K}|. \tag{2.10}$$

This can be shown by a recursive argument on the number of $U(1)$ factors.

**Index for the non-abelian theory.** Let us denote the Witten index of the general unitary CS theory (2.1)-(2.2) by:

$$\mathbf{I}_W[\boldsymbol{N}\,\boldsymbol{l}\,|\,\boldsymbol{k}] \equiv \mathbf{I}_W\begin{bmatrix} N_1 & l_1 & k_1 & k_{12} & \cdots & k_{1n} \\ N_2 & l_2 & k_{12} & k_2 & \cdots & k_{2n} \\ \vdots & \vdots & \vdots & \vdots & \ddots & \vdots \\ N_n & l_n & k_{1n} & k_{2n} & \cdots & k_n \end{bmatrix}. \tag{2.11}$$

Then, given the above observations, we find that:

$$\mathbf{I}_W[\boldsymbol{N}\,\boldsymbol{l}\,|\,\boldsymbol{k}] = |\det \boldsymbol{K}|\prod_{i=1}^{n}\frac{1}{N_i^2}\binom{|k_i|-1}{N_i-1}, \tag{2.12}$$

where $\boldsymbol{K}$ is defined as in (2.7). Of course, this reduces to (2.5) for $n=1$.

## 2.2 Generalised $\mathcal{N}=2$ level/rank dualities

The $U(N)_{k,k+lN}$ $\mathcal{N}=2$ CS theory has a dual description [18]:

$$U(N)_{k,k+lN} \qquad \longleftrightarrow \qquad U(|k|-N)_{-k,-k+\epsilon(k-N)} \times U(1)_{l+\epsilon}, \tag{2.13}$$
$$\underbrace{\phantom{U(|k|-N)_{-k,-k+\epsilon(k-N)} \times U(1)_{l+\epsilon}}}_{\epsilon}$$

with $\epsilon \equiv \text{sign}(k)$. Here the FI parameter on the electric (left-hand) side maps to an FI parameter for the $U(1)_{l+\epsilon}$ factor on the magnetic (right-hand) side. In addition, on the magnetic side, we also have the non-zero CS contact terms:

$$K_{RR} = \begin{cases} -(k-N)^2, & \text{if } \epsilon = 1, \\ N^2 - 1, & \text{if } \epsilon = -1, \end{cases} \qquad K_g = \begin{cases} -2k(k-N), & \text{if } \epsilon = 1, \\ -2kN - 2, & \text{if } \epsilon = -1. \end{cases} \tag{2.14}$$

This is a special case of the Nii duality [18] which we further studied in [5]. It generalises well-known level/rank dualities for pure CS theories [27], as can be shown by writing these dualities in $\mathcal{N} = 0$ language – for completeness, we discuss this in appendix A. Note that for $N = |k|$, the dual description is abelian:

$$U(|k|)_{k,k+l|k|} \qquad \longleftrightarrow \qquad U(1)_{l+\epsilon}. \tag{2.15}$$

Another interesting special case is for $l = -\epsilon$, in which case the duality can be simplified to:

$$U(N_c)_{k,k+lN} \qquad \longleftrightarrow \qquad SU(|k|-N)_{-k}. \tag{2.16}$$

Correspondingly, the Witten index (2.5) specialises to:

$$\mathbf{I}_W [N - \epsilon \,|\, k] = \binom{|k|-1}{N^D - 1} = \mathbf{I}_W \left[ SU(N^D)_{-k} \right], \tag{2.17}$$

with $N^D = |k| - N$.

**An almost trivial $U(1) \times U(1)$ theory.** Consider the abelian theory:

$$\underbrace{U(1)_{l+\epsilon} \times U(1)_{l+\epsilon'}}_{l}, \tag{2.18}$$

with $\epsilon, \epsilon'$ equal to 1 or $-1$, and $l \in \mathbb{Z}$. The Witten index is given by

$$\mathbf{I}_W = |(l + \epsilon)(l + \epsilon') - l^2| = 1, \tag{2.19}$$

where the last equality holds if $\epsilon = -\epsilon'$, as we assume in the following. Then we have a unique Bethe vacuum. The Bethe equations read $q(x')^l(-x)^{l+\epsilon} = 1$, $q'x^l(-x')^{l+\epsilon'} = 1$, with the unique Bethe vacuum given by:

$$x = (-q)^{l+\epsilon'}(q')^{-l}, \qquad x' = q^{-l}(-q')^{l+\epsilon}. \tag{2.20}$$

Here $x, x'$ and $q, q'$ denote the gauge parameter and the topological symmetry fugacities $U(1)_T \times U(1)_{T'}$, respectively. Using the 3d $A$-model, we can derive (most of) the CS contact terms that remain in the dual description. This gives us the elementary duality:

$$\underbrace{U(1)_{l+\epsilon} \times U(1)_{l+\epsilon'}}_{l} \qquad \longleftrightarrow \qquad \begin{cases} K_{TT} = l + \epsilon', & K_{T'T'} = l + \epsilon, \\ K_{TT'} = -l, & K_{RR} = 1, \end{cases} \tag{2.21}$$

up to some gravitational CS level $K_g$ which we did not fully determine (see appendix A). For $l = 0$, this gives us two decoupled elementary dualities for $U(1)_{\pm 1}$ (see [5] for a recent review).

**An $U(N) \times U(N')$ level/rank duality.** The level/rank duality (2.13) has an interesting generalisation for a gauge group $U(N) \times U(N')$. Setting $\epsilon = \text{sign}(k)$, $\epsilon' = \text{sign}(k')$ and assuming $\epsilon = -\epsilon'$, we find:

$$\underbrace{U(N)_{k,k+lN} \times U(N')_{k',k'+lN'}}_{l}$$
$$\longleftrightarrow \quad \underbrace{U(|k|-N)_{-k,-k+l(|k|-N)} \times U(|k'|-N')_{-k',-k'+l(|k'|-N')}}_{l}, \tag{2.22}$$

where the FI parameters of the dual theory are related to the original FI parameters $\tau, \tau'$ as:

$$\tau_D = -\tau + l\epsilon(\tau - \tau'), \qquad \tau'_D = -\tau' + l\epsilon'(\tau' - \tau), \qquad (2.23)$$

as well the following CS contact terms for the topological symmetries:[2]

$$K_{TT} = l + \epsilon', \qquad K_{T'T'} = l + \epsilon, \qquad K_{TT'} = -l. \qquad (2.24)$$

This duality also implies the equality:

$$\mathbf{I}_W \begin{bmatrix} N_c & l & k & l \\ N'_c & l & l & k' \end{bmatrix} = \mathbf{I}_W \begin{bmatrix} |k| - N_c & l & -k & l \\ |k'| - N'_c & l & l & -k' \end{bmatrix}, \qquad (2.25)$$

if $\epsilon + \epsilon' = 0$. This indeed follows from the explicit expression (2.12) for the Witten index.

The duality (2.22) can be derived using (2.13) and (2.21). Indeed, applying the duality (2.13) subsequently to each gauge group factor on the electric side of (2.22), one finds:

$$\underbrace{U(|k| - N_c)_{-k, -k+\epsilon(|k|-N_c)}}_{\epsilon} \times \underbrace{U(1)_{l+\epsilon} \times U(1)_{l+\epsilon'}}_{l} \times \underbrace{U(|k'| - N'_c)_{-k', -k'+\epsilon'(|k'|-N'_c)}}_{\epsilon'}. \qquad (2.26)$$

The $U(1) \times U(1)$ sector can be simplified thanks to the elementary duality (2.21). Due to the mixed CS couplings, the effective FI parameters for the $U(1) \times U(1)$ gauge group are:

$$\tau_{\text{eff}} = \tau + \epsilon \sum_{a=1}^{|k|-N} u_a, \qquad \tau'_{\text{eff}} = \tau' + \epsilon' \sum_{a=1}^{|k'|-N'} u'_a, \qquad (2.27)$$

where $u_a, u'_a$ denote the gauge parameters of the non-abelian factors in (2.26). Applying the duality (2.21) then leads to (2.22). It is simplest to check this using the 3d $A$-model – that is, on the 2d Coulomb branch of the 2d $\mathcal{N} = (2,2)$ field theory description obtained by circle compactification [5]. Let us denote by $u$ and $u'$ the sums in (2.27), so that $\tau_{\text{eff}} = \tau + \epsilon u$ and $\tau'_{\text{eff}} = \tau' + \epsilon' u'$. Then the relevant part of the twisted superpotential reads:[3]

$$\mathcal{W} = \frac{\epsilon}{2} u^2 + \frac{\epsilon}{2} u^2 + \mathcal{W}_{\text{magn}}(u, u'), \qquad (2.28)$$

where the first two terms are the '$l$ CS levels' for the non-abelian factors in (2.26), and $\mathcal{W}_{\text{magn}}(u, u')$ is the superpotential for the abelian factor after applying (2.21), which reads:

$$\mathcal{W}_{\text{magn}}(u, u') = \frac{l + \epsilon'}{2}(\tau + \epsilon u)^2 + \frac{l + \epsilon}{2}(\tau' + \epsilon' u')^2 - l(\tau + \epsilon u)(\tau' + \epsilon' u'). \qquad (2.29)$$

Expanding out (2.28) and using $\epsilon\epsilon' = -1$, we recover the CS levels shown in (2.22) and (2.24).

## 3 Witten index and semi-classical vacua of unitary SQCD

In this section, we further comment on the Witten index of the $\mathcal{N} = 2$ SQCD theory that we studied in [5]. We show that the Witten index for an arbitrary number of fundamental and antifundamental chiral multiples can be derived from the Witten index for SQCD with only fundamental multiplets, which we will study thoroughly in section 4.

---

[2]There are also non-zero contact terms $K_{RR}$ and $K_g$, which we do not keep track of here.

[3]Here, for simplicity of notation, we omit the terms linear in $u$, $u'$ that arise from the $U(1)$ CS levels [35].

Unitary SQCD$[N_c, k, l, n_f, n_a]$ is the 3d $\mathcal{N} = 2$ gauge theory with the gauge group $U(N_c)_{k, k+lN_c}$ coupled to $n_f$ fundamental chiral multiplets and to $n_a$ antifundamental chiral multiplets. Its physics is governed, in part, by the 'chirality parameter':

$$k_c \equiv \frac{1}{2}(n_f - n_a). \tag{3.1}$$

We have $k + k_c \in \mathbb{Z}$ and $l \in \mathbb{Z}$ for consistency – see [5] for more details on our conventions. In [5], we computed the Witten index in the case $n_f = n_a \equiv N_f$:

$$\mathbf{I}_W[N_c, k, l, N_f, N_f] = \begin{cases} \frac{N_f + |l|N_c}{N_f} \binom{N_f}{N_c}, & \text{if } k = 0, \\ \sum_{j=0}^{N_f} \frac{|k + l(N_c - j)|}{|k|} \binom{N_f}{j} \binom{|k|}{N_c - j}, & \text{if } k \neq 0. \end{cases} \tag{3.2}$$

For $N_f = 0$ (with $k \neq 0$, $N_c = N$), this reduces to (2.5).

## 3.1 Semiclassical vacuum moduli space

The flavoured Witten index (1.2) of any 3d $\mathcal{N} = 2$ supersymmetric gauge theory captures the number of vacua of that theory deformed by real masses $m = -\frac{1}{2\pi} \log |y|$ associated with the flavour symmetry. Whenever the mass deformation is generic enough to lift all non-compact branches of the vacuum moduli space, the Witten index is well-defined. Moreover, it is independent of the specific mass deformation chosen [3].

Let $Q_i^a$ and $\widetilde{Q}_a^j$ denote the fundamental and antifundamental scalars in the chiral multiplets of SQCD$[N_c, k, l, n_f, n_a]$. With generic mass deformations (including the FI parameter $\xi$) and upon diagonalising the real adjoint scalar, $\sigma \rightarrow \text{diag}(\sigma_a)$, the semi-classical vacuum equations read:

$$\begin{aligned}
(\sigma_a - m_i) Q_i^a &= 0, & i &= 1, \cdots, n_f, \\
(-\sigma_a + \widetilde{m}_j) \widetilde{Q}_a^j &= 0, & j &= 1, \cdots, n_a, \\
\sum_{i=1}^{n_f} Q_a^{\dagger i} Q_i^b - \sum_{j=1}^{n_a} \widetilde{Q}_j^{\dagger b} \widetilde{Q}_a^j &= \frac{\delta_a{}^b}{2\pi} F_a(\sigma; \xi, m),
\end{aligned} \tag{3.3}$$

with $a = 1, \cdots, N_c$ (and no summation implied). The piecewise-linear functions $F_a$ read:

$$F_a(\sigma; \xi, m) = \xi + k\sigma_a + l \sum_{b=1}^{N_c} \sigma_b + \frac{1}{2} \sum_{i=1}^{n_f} |\sigma_a - m_i| - \frac{1}{2} \sum_{j=1}^{n_a} |-\sigma_a + \widetilde{m}_j|. \tag{3.4}$$

These include the contribution from one-loop shifts to the effective CS levels and FI parameter [3]. Here we use a convenient notation where $m_i$, $\widetilde{m}_j$ are $U(n_f) \times U(n_a)$ parameters, which includes part of the gauge symmetry – the actual flavour symmetry being $U(n_f) \times U(n_a)/U(1) \cong SU(n_f) \times SU(n_a) \times U(1)_A$, plus the topological symmetry $U(1)_T$.[4]

Solving the vacuum equations (3.3)-(3.4) with generic mass parameters, we expect to find only discrete solutions – in that case, the number of vacua would give the Witten index (1.2). This coupled system of equations is closely related to the Bethe equations studied in [5], and it is similarly complicated. For our purpose, it will be more useful to consider particular limits on the masses such that the index remains well-defined. Doing this, we will encounter three types of (semi-classical) vacua:

---

[4]Here, we are not being particularly careful about the global form of the flavour group. In the case $n_a = 0$, on which we focus in this work, the full $SU(n_f)$ group acts as a symmetry.

(i) The *Higgs vacua* are compact moduli spaces $\mathcal{M}_H$, including the case of discrete vacua. Any such compact branch $\mathcal{M}_H$ contributes to the Witten index through its Euler characteristic,

$$\chi(\mathcal{M}_H) \subset \mathbf{I}_W. \tag{3.5}$$

In particular, we will often encounter $\mathcal{M}_H$ the Grassmannian manifold $\mathrm{Gr}(p, n)$, which contributes a binomial coefficient:

$$\mathbf{I}_W\left[\mathrm{Gr}(p, n)\right] = \chi\left(\mathrm{Gr}(p, n)\right) = \binom{n}{p}. \tag{3.6}$$

Higgs vacua correspond to having the scalars $Q$, $\widetilde{Q}$ non-vanishing.

(ii) The *topological vacua* consist of pure $\mathcal{N} = 2$ supersymmetric Chern-Simons theories; this is equivalent to having a 3d topological quantum field theory (TQFT) in the infrared, as the gauginos are massive. In this paper, we will only encounter CS theories with unitary gauge groups. Such TQFT sectors contribute to the Witten index through the non-zero index of the $\mathcal{N} = 2$ CS theory,

$$\mathbf{I}_W[\mathrm{TQFT}] \subset \mathbf{I}, \tag{3.7}$$

which we studied in section 2. Topological vacua arise at fixed non-zero values of the fields $\sigma_a$ so that all chiral multiplets are massive and can be integrated out, leaving behind an effective pure CS theory.

(iii) The *Coulomb vacua* are semi-classical vacua that open up where (part of) the non-abelian gauge symmetry is restored (with vanishing effective CS levels). In this case, we have continuous solutions for $\sigma_a \neq 0$ and the semi-classical analysis is usually not reliable – some strong-coupling effect may modify the picture entirely [36]. Hence, when such vacua arise, we will have to make some *ad-hoc* conjectures about the corresponding contribution to the index:

$$\mathbf{I}_W[\text{strongly-coupled vacua}] \subset \mathbf{I}_W. \tag{3.8}$$

Moreover, in general we may have *hybrid vacua* where some of these possibilities arise simultaneously. In particular we will find many *Higgs-topological vacua*. These are simply cases where there is a residual TQFT at every point on the Higgs branch, which we may view as a trivial fibration $\mathrm{TQFT} \to \mathcal{M}_{\mathrm{hybrid}} \to \mathcal{M}_H$.[5] We denote such vacua by $\mathcal{M}_H \times \mathrm{TQFT}$. Their contribution to the Witten index is simply the product of the geometric and TQFT contributions,

$$\mathbf{I}_W[\mathcal{M}_{\mathrm{hybrid}}] = \chi(\mathcal{M}_H)\mathbf{I}_W[\mathrm{TQFT}] \subset \mathbf{I}_W. \tag{3.9}$$

We will discuss all these cases more thoroughly in the next section.

## 3.2 $U(1)_k$ with $n_f$ chiral multiplets of charge $+1$: a review

Before tackling the non-abelian case, it will be useful to review the computation of the index for a $U(1)_k$ gauge theory coupled to $n_f$ chiral multiplets of charges $+1$ [3]. Recall that the CS level $k$ is quantised as $k + \frac{n_f}{2} \in \mathbb{Z}$. We consider the case with zero mass for the chiral

---

[5]In a general gauge theory, we could have a non-trivial fibration because different subgroups of the gauge group might survive at different points on the Higgs branch. Here, at each solution for the $\sigma$'s and $Q$'s, we have a standard Higgs mechanism and the TQFT arises from gauge fields that do not couple at all to the chiral multiplets that obtain a VEV, hence the fibration is trivial.

multiplet, but we assume that the FI parameter is non-zero. In that case, the semi-classical vacuum equations (3.3) become:

$$\sigma Q_i = 0, \qquad\qquad i = 1, \cdots, n_f,$$
$$\sum_{i=1}^{n_f} Q_i^\dagger Q_i = \frac{1}{2\pi} F(\sigma), \qquad F(\sigma) \equiv \xi + k\sigma + \frac{n_f}{2}|\sigma|. \tag{3.10}$$

**Higgs vacua:** The first equation in (3.10) implies that Higgs vacua may only arise at the origin of the would-be Coulomb branch ($\sigma = 0$). Then, the Higgs branch is governed by the standard $D$-term equation:

$$\sum_{i=1}^{n_f} Q_i^\dagger Q_i = \xi. \tag{3.11}$$

There are no solutions for $\xi < 0$, so let us assume that $\xi > 0$, for now. Then, upon quotienting by the $U(1)$ gauge group, we have a standard Kähler quotient description of the projective space $\mathbb{CP}^{n_f-1}$. Its contribution to the total Witten index is thus:

$$\mathbf{I}_{W,\mathrm{I}}\big[1, k, 0, n_f, 0\big] = \chi\big(\mathbb{CP}^{n_f-1}\big) = n_f. \tag{3.12}$$

In anticipation of the next section, we call this a solution of *type I*. Note that this solution exists for any value of $k$.

**Topological vacua:** These arise if $Q = 0$ and $\sigma \neq 0$, in which case we need to find non-trivial solutions to the equation:

$$F(\sigma) = \xi + \left(k + \mathrm{sign}(\sigma)\frac{n_f}{2}\right)\sigma = 0. \tag{3.13}$$

We shall call such solutions the *type III* solutions.[6] Let us start with the non-marginal case, that is with $|k| \neq \frac{n_f}{2}$. Assuming that $\xi > 0$, we have two solutions (depending on the sign of $\sigma$):

$$\begin{cases} \sigma^+ = -\frac{\xi}{k+\frac{n_f}{2}} > 0, & \text{iff } k + \frac{n_f}{2} < 0, \\ \sigma^- = -\frac{\xi}{k-\frac{n_f}{2}} < 0, & \text{iff } k - \frac{n_f}{2} > 0. \end{cases} \tag{3.14}$$

We will use the notation $\sigma^\pm$ for solutions for $\sigma$ such that $\sigma^+ > 0$ and $\sigma^- < 0$, respectively. We then have the following effective abelian CS theories:

$$\mathcal{M}_{\mathrm{III}}^{\xi>0}\big[1, k, 0, n_f, 0\big] = \Theta\left(-k - \frac{n_f}{2}\right) U(1)_{k+\frac{n_f}{2}} \oplus \Theta\left(k - \frac{n_f}{2}\right) U(1)_{k-\frac{n_f}{2}}. \tag{3.15}$$

Here and in the following, we find it convenient to use the Heaviside step function:

$$\Theta(x) \equiv \begin{cases} 1, & \text{if } x > 0, \\ 0, & \text{if } x \leq 0, \end{cases} \tag{3.16}$$

to keep track of constraints on the parameters. (In the present case, we have a TQFT contribution $U(1)_{k\pm\frac{n_f}{2}}$ for $k < -\frac{n_f}{2}$ and $k > \frac{n_f}{2}$, respectively, and no TQFT if $|k| < \frac{n_f}{2}$. For $|k| = \frac{n_f}{2}$, there is a naive $U(1)_0$ contribution which is lifted by the FI term.) As discussed around (2.8), the Witten index for the $U(1)_K$ CS theory is equal to $|K|$, hence the vacua (3.15) contribute:

$$\mathbf{I}_{W,\mathrm{III}}^{\xi>0}\big[1, k, 0, n_f, 0\big] = \begin{cases} |k| - \frac{n_f}{2}, & \text{if } |k| > \frac{n_f}{2}, \\ 0, & \text{if } |k| < \frac{n_f}{2}, \end{cases} \tag{3.17}$$

---

[6]In the non-abelian case, we will also find Higgs-topological vacua, which will be the *type II* solutions.

to the total index of the abelian theory. We must also carefully treat the marginal cases, $k = \pm\frac{n_f}{2}$, in which case we find the solutions

$$\sigma^\pm = \mp\frac{\xi}{n_f}, \qquad \text{iff } \xi < 0. \tag{3.18}$$

Hence there are no topological solutions for $\xi > 0$.

**The full Witten index.** Adding the contributions (3.12) and (3.17) in the regime $\xi > 0$, one finds:

$$\mathbf{I}_W\left[U(1)_k \oplus n_f \,\Box\right] = \begin{cases} |k| + \frac{n_f}{2}, & \text{if } |k| \geq \frac{n_f}{2}, \\ n_f, & \text{if } |k| < \frac{n_f}{2}. \end{cases} \tag{3.19}$$

As already mentioned, the index is independent of the mass deformation as long as all non-compact directions of the moduli space are lifted. If we now consider the case $\xi < 0$, there are no Higgs vacua but we find the following topological vacua:

$$\begin{cases} \sigma^+ = -\frac{\xi}{k+\frac{n_f}{2}} > 0, & \text{iff } k + \frac{n_f}{2} > 0, \\ \sigma^- = -\frac{\xi}{k-\frac{n_f}{2}} < 0, & \text{iff } k - \frac{n_f}{2} < 0, \end{cases} \tag{3.20}$$

as well as the vacua (3.18) in the marginal cases. Hence we have:

$$\mathcal{M}_{\text{III}}^{\xi<0}\left[1,k,0,n_f,0\right] = \Theta\left(k+\frac{n_f}{2}\right)U(1)_{k+\frac{n_f}{2}} \oplus \Theta\left(-k+\frac{n_f}{2}\right)U(1)_{k-\frac{n_f}{2}}, \tag{3.21}$$

which indeed reproduces (3.19).

## 3.3 Recursion relation for the flavoured Witten index

Let us now consider the Witten index of SQCD$[N_c, k, l, n_f, n_a]$ with $n_f \geq n_a$, without loss of generality.[7] The index with $n_f = n_a$ is given by (3.2), hence we shall focus on the 'chiral' case $n_f > n_a$ (that is, $k_c > 0$). We will show that the Witten index satisfies the recursion relation:

$$\mathbf{I}_W[N_c, k, l, n_f, n_a] = \mathbf{I}_W[N_c, k, l, n_f - 1, n_a - 1] + \mathbf{I}_W[N_c - 1, k, l, n_f - 1, n_a - 1]. \tag{3.22}$$

**Derivation.** Assuming that $n_a > 0$, consider the $U(1)_{n_f} \times U(1)_{n_a} \subset U(n_f) \times U(n_a)$ flavour symmetry under which the chiral multiplets $Q_{n_f}$ and $\widetilde{Q}^{n_a}$ are charged, and let us turn on a large mass $m$ for the diagonal $U(1) \subset U(1)_{n_f} \times U(1)_{n_a}$. Then, the vacuum equations take the form:

$$\begin{aligned} &(\sigma_a - m)Q_{n_f}^a = 0, && (-\sigma_a + m)\widetilde{Q}_a^{n_a} = 0, \\ &\sigma_a Q_i^a = 0, \quad i = 1, \cdots, n_f - 1, && \sigma_a \widetilde{Q}_a^j = 0, \quad j = 1, \cdots, n_a - 1, \\ &\sum_{i=1}^{n_f} Q_a^{\dagger i} Q_i^b - \sum_{j=1}^{n_a} \widetilde{Q}_j^{\dagger b} \widetilde{Q}_a^j = \frac{\delta_a{}^b}{2\pi}F_a(\sigma), && F_a(\sigma, m) \equiv \xi + (k + \text{sign}(\sigma_a)k_c)\sigma_a + l\sum_{b=1}^{N_c}\sigma_b, \end{aligned} \tag{3.23}$$

with $k_c \equiv \frac{1}{2}(n_f - n_a)$. More precisely, we assume that $|m_i|, |\widetilde{m}_j| \ll |m|$ for $i \neq n_f$, $j \neq n_a$, so we may ignore those small masses for simplicity of notation. In this limit, the solutions decomposes into two disjoint families. Firstly, we have the solutions with $|\sigma_a| \ll |m|$, in which case the vacuum equations correspond to the low-energy effective field theory:

$$U(N_c)_{k, k+lN_c} \quad \text{coupled with} \quad (n_f - 1)\,\Box \oplus (n_a - 1)\,\overline{\Box}. \tag{3.24}$$

---

[7]This also gives us the result for $n_a > n_f$ since $\mathbf{I}_W[N_c, k, l, n_f, n_a] = \mathbf{I}_W[N_c, k, l, n_a, n_f]$.

Indeed, the large real mass $m$ allows us to integrate out the flavour pair $Q_{n_f}$, $\widetilde{Q}^{n_a}$, and the CS levels do not change in that particular limit. Secondly, we should consider potential solutions with $\sigma_{N_c} \approx m$. In that limit, we Higgs the gauge group to:

$$U(N_c) \longrightarrow \underbrace{U(N_c - 1)_{k, k+lN_c}}_{l} \times U(1)_{k_{\text{eff}}}, \tag{3.25}$$

At low energy, there are still $n_f - 1$ fundamental chiral multiplets and $n_a - 1$ antifundamental chiral multiplets charged under the $U(N_c - 1)$ factor, while the chiral multiplets $Q_{n_f}^a$, $\widetilde{Q}_a^{n_a}$ ($a < N_c$) charged under $U(N_c - 1)$ are integrated out as above. On the other hand, the fields $Q_{n_f}^{N_c}$, $\widetilde{Q}_{N_c}^{n_a}$ remain light, thus contributing one flavour (i.e. two chiral multiplets of charge $\pm 1$) to the abelian sector. Moreover, integrating out all the massive fields leads to a shift of the abelian CS level according to:

$$k_{\text{eff}} = k + l + \text{sign}(\sigma_{N_c})k_c. \tag{3.26}$$

We have $\sigma_{N_c} = m + \delta$, with $\delta$ very small by assumption. Naively, we might think that the $U(1)$ sector in (3.25) would contribute non-trivially to the Witten index, as the Witten index of a $U(1)_{k_{\text{eff}}}$ theory with one flavour is $1 + |k_{\text{eff}}|$. However, this includes $|k_{\text{eff}}|$ topological vacua which would arise at parametrically large values of $\delta$; this would violate our assumption and therefore we should not count those putative vacua [3]. As far as computing the Witten index is concerned, therefore, the second class of solutions is isomorphic to the number of vacua for the partially Higgsed theory:

$$U(N_c - 1)_{k, k+lN_c} \quad \text{coupled with} \quad (n_f - 1)\,\square \oplus (n_a - 1)\,\overline{\square}. \tag{3.27}$$

In this way, we just derived the advertised recursion relation (3.22). It can be also be derived, completely analogously, by looking at particular limits of the Bethe equations [5,37].

**Witten index: the general case.** Using the recursion relation (3.22) and assuming that $n_f > n_a$, we find the following explicit expression for the index:

$$\mathbf{I}_W[N_c, k, l, n_f, n_a] = \sum_{j=0}^{n_a} \binom{n_a}{j} \mathbf{I}_W[N_c - j, k, l, n_f - n_a, 0]. \tag{3.28}$$

Thus, to compute $\mathbf{I}_W[N_c, k, l, n_f, n_a]$ in general, all we have left to do is to explicitly compute $\mathbf{I}_W[N_c, k, l, n_f, 0]$, namely the Witten index for 'chiral' SQCD with only fundamental matter. This is the theory we will study in the rest of this paper.

## 4 Witten Index for $U(N_c)_{k, k+lN_c}$ with $n_f$ fundamentals

Given the previous discussion, we now focus on the $U(N_c)_{k, k+lN_c}$ gauge theory with $n_f$ fundamental chiral multiplets. Setting the masses to zero and turning on a non-zero FI term, the semi-classical vacuum equations (3.3)-(3.4) reduce to:

$$\begin{aligned}
\sigma_a Q_i^a &= 0, & i &= 1, \cdots, n_f, \\
\sum_{i=1}^{n_f} Q_a^{\dagger i} Q_i^b &= \frac{\delta_a{}^b}{2\pi} F_a(\sigma), & F_a(\sigma) &= \xi + k\sigma_a + l\sum_{b=1}^{N_c} \sigma_b + \frac{n_f}{2}|\sigma_a|,
\end{aligned} \tag{4.1}$$

with $a = 1, \cdots, N_c$. In the abelian case ($N_c = 1$), the solutions were discussed in section 3.2. As we will now show, the structure of the vacuum with $\xi \neq 0$ in the non-abelian case is rather

intricate, especially for non-zero values of $l$. Let us first explain the general structure of the solution, before discussing it in more details – to be pedagogical, we will first analyse the $U(2)$ theory in subsection 4.1 before giving the general $U(N_c)$ result in subsection 4.2.

**Higgs vacuum (*Type I*): the complex Grassmannian.** The simplest solutions are for $\sigma_a = 0$, $\forall a$, which we call the *Type I* vacua. This corresponds to the solution to the $D$-term relation:

$$\sum_{i=1}^{n_f} Q_a^{\dagger i} Q_i^b = \frac{\delta_a{}^b}{2\pi} \xi \,, \tag{4.2}$$

modulo $U(N_c)$ gauge transformations. For $\xi > 0$, this famously gives us the complex Grassmannian manifold $\mathrm{Gr}(N_c, n_f)$ (see *e.g.* [28]), while there are no solution for $\xi < 0$. We write this vacuum and its contribution to the Witten index as:

$$\mathcal{M}_{\mathrm{I}} = \Theta(\xi) \, \mathrm{Gr}(N_c, n_f) \,, \qquad \mathbf{I}_{W,\mathrm{I}} = \Theta(\xi) \begin{pmatrix} n_f \\ N_c \end{pmatrix} . \tag{4.3}$$

For general values of the CS levels $k$, $l$, there will be many more topological and Higgs-topological vacua, as well as some strongly-coupled vacua. As already mentioned in [5], the geometric contribution (4.3) provides a lower bound for the Witten index. This is because all other vacua that we find (at $\xi > 0$) are bosonic, as we will see, so they all contribute to the index with a positive sign.

**Higgs-topological (*Type II*) and topological vacua (*Type III*).** All such vacua arise as solutions with $\sigma_a \neq 0$ for at least some $\sigma_a$'s. Due to the residual gauge transformations that permutes the $\sigma$'s, we only need to specify how many $\sigma$'s are zero and how many are non-zero. We encounter the following four possibilities:

*Type II,a*: What we call Type II vacua are the solutions such that some but not all $\sigma$'s vanish. We then pick $\sigma_a = 0$ for $a = 1, \cdots, N_c - p$, and $\sigma_{a'} \neq 0$ for $a' = N_c - p + 1, \cdots, N_c$, for $0 < p < N_c$. In the *Type II,a* case, we choose the $\sigma_{a'}$'s to all have the same sign. We then obtain a hybrid Higgs-topological vacuum $\mathrm{Gr}(N_c - p) \times U(p)$, where $U(p)$ denotes some pure CS theory with gauge group $U(p)$.

*Type II,b*: These are the Type II vacua such that, out of $p + q > 0$ non-zero $\sigma$'s, we choose $p$ of them to be positive and $q$ of them to be negative, which would lead to a Higgs-topological vacuum $\mathrm{Gr}(N_c - p - q) \times U(p) \times U(q)$. In the end, it will turn out that there are no such vacua in our theory. We should note that some of these 'Higgs-topological vacua' are actually ordinary Higgs vacua – this occurs whenever the TQFT sector is (dual to) a trivial theory with a single state.

*Type III,a*: What we call Type III vacua are topological vacua, which arise when all the $\sigma$'s are non-zero. *Type III,a* vacua corresponds to choosing all $\sigma_a$'s to have the same sign, in which case we obtain an effective $U(N_c)$ CS theory.

*Type III,b*: For these vacua, we choose $p$ of the $\sigma$'s to be positive, and $N_c - p$ of them to be negative, which then leaves us with an effective $U(p) \times U(N_c - p)$ CS theory.

**Strongly-coupled vacua (*Type IV*).** Finally, we have to address the logical possibility that there might exist strongly-coupled vacua at small values of $\sigma$. By comparing our computation to the Bethe-vacua counting of [5], we find that such vacua arise whenever there exists *Coulomb vacua* in the semi-classical analysis (see section 3.1). Each such semi-classical Coulomb branch corresponds to setting $N_c - p$ $\sigma$'s to zero, and having an effective CS level 0 for some $SU(p) \subset U(p) \subseteq U(N_c)$ unhiggsed subgroup that may appear at some values of the

$\sigma$'s. The effective 3d $\mathcal{N} = 2\, SU(p)_0$ gauge theory without matter does not have any stable vacuum [36]. However, we expect that, in those cases, there exists additional, strongly-coupled supersymmetric vacua that survive near the origin of the Coulomb branch (which is otherwise lifted non-perturbatively). These 'true Type IV' vacua are not captured by our analysis, but we are nonetheless able to derive their contribution to the index in various ways (in particular by computing the index for either sign of the FI parameter; see also section 5.3 for the dual perspective).

### 4.1 $U(2)_{k,k+2l}$ with $n_f$ fundamentals

Let us first consider the $U(2)_{k,k+2l}$ gauge theory with $n_f$ fundamentals. We wish to solve the vacuum equations:

$$
\begin{aligned}
\sigma_a Q_i^a &= 0, & a &= 1, 2, \quad i = 1, \cdots, n_f, \\
\sum_{i=1}^{n_f} Q_a^{\dagger i} Q_i^b &= \frac{\delta_a^{\ b}}{2\pi} F_a(\sigma), & F_a(\sigma) &= \xi + k\sigma_a + l(\sigma_1 + \sigma_2) + \frac{n_f}{2}|\sigma_a|.
\end{aligned}
\tag{4.4}
$$

Let us now consider all possible solutions to these equations. The **Type I** solution is as discussed above – we have a Grassmannian manifold $\mathrm{Gr}(2, n_f)$ if $\xi > 0$, which contributes:

$$
\mathbf{I}_{W,\mathrm{I}}[2, k, l, n_f, 0] = \Theta(\xi) \binom{n_f}{2} = \Theta(\xi) \frac{n_f(n_f - 1)}{2}.
\tag{4.5}
$$

There are also many possible Higgs-topological and topological vacua, as well as potential strongly-coupled vacua if $|k| = \frac{n_f}{2}$, as we will now discuss.

**Type II vacua.** If we assume that $\sigma_1 = 0$ and $\sigma_2 \neq 0$ in (4.4), we have to solve the equations:

$$
2\pi \sum_{i=1}^{n_f} Q_1^{\dagger i} Q_i^1 = F_1(\sigma) = \xi + l\sigma_2 > 0, \qquad F_2(\sigma) = \xi + \left(k + l + \mathrm{sign}(\sigma_2)\frac{n_f}{2}\right)\sigma_2 = 0. \tag{4.6}
$$

Note the inequality $F_1(\sigma) > 0$, which is necessary for the Higgs-branch $\mathbb{CP}^{n_f-1}$ to exists, while the solutions to $F_2(\sigma) = 0$ correspond to TQFTs ($U(1)$ CS theories). Let us first observe that we cannot obtain any solution in the case $k + l + \mathrm{sign}(\sigma_1)\frac{n_f}{2} = 0$, since $\xi \neq 0$ by assumption. (Similar cases will appear repeatedly in the following analysis, and we will not discuss them explicitly.) Now, at non-zero values on the real line $\sigma_2$, we have two solutions analogous to (3.14), namely:

$$
\begin{cases}
\sigma_2^+ = -\dfrac{\xi}{k+l+\frac{n_f}{2}} > 0, & \text{iff } \mathrm{sign}(\xi)\left(k+l+\frac{n_f}{2}\right) < 0 \text{ and } k + \frac{n_f}{2} < 0, \\[2ex]
\sigma_2^- = -\dfrac{\xi}{k+l-\frac{n_f}{2}} < 0, & \text{iff } \mathrm{sign}(\xi)\left(k+l-\frac{n_f}{2}\right) > 0 \text{ and } k - \frac{n_f}{2} > 0,
\end{cases}
\tag{4.7}
$$

where the other inequalities above arise from demanding the volume of the Higgs branch to be positive.

This gives us the second component of the moduli space of vacua of the $U(2)$ theory:

$$
\begin{aligned}
\mathcal{M}_{\mathrm{II}}[2, k, l, n_f, 0] &= \Theta\left(-k - \frac{n_f}{2}\right)\Theta\left(\xi\left(-k - l - \frac{n_f}{2}\right)\right)\mathbb{CP}^{n_f-1} \times U(1)_{k+l+\frac{n_f}{2}} \\
&\oplus \Theta\left(k - \frac{n_f}{2}\right)\Theta\left(\xi\left(k + l - \frac{n_f}{2}\right)\right)\mathbb{CP}^{n_f-1} \times U(1)_{k+l-\frac{n_f}{2}},
\end{aligned}
\tag{4.8}
$$

which contributes to the Witten index as:

$$
\begin{aligned}
\mathbf{I}_{W,\mathrm{II}}[2, k, l, n_f, 0] &= \Theta\left(-k - \frac{n_f}{2}\right)\Theta\left(\xi\left(-k - l - \frac{n_f}{2}\right)\right) n_f \left|k + l + \frac{n_f}{2}\right| \\
&+ \Theta\left(k - \frac{n_f}{2}\right)\Theta\left(\xi\left(k + l - \frac{n_f}{2}\right)\right) n_f \left|k + l - \frac{n_f}{2}\right|.
\end{aligned}
\tag{4.9}
$$

**Type III vacua.** Next, we consider the topological vacua. For the Type III,a solutions, we choose $\sigma_1, \sigma_2$ to be of the same sign. It then turns out that $\sigma_1 = \sigma_2$, and we find the two solutions:

$$\begin{cases} \sigma_1 = \sigma_2 = \sigma^+ \equiv -\dfrac{\xi}{k+2\ell+\frac{n_f}{2}} > 0, & \text{if } \operatorname{sign}(\xi)\left(k+2\ell+\frac{n_f}{2}\right) < 0, \\[2mm] \sigma_1 = \sigma_2 = \sigma^- \equiv -\dfrac{\xi}{k+2\ell-\frac{n_f}{2}} < 0, & \text{if } \operatorname{sign}(\xi)\left(k+2\ell-\frac{n_f}{2}\right) > 0, \end{cases} \tag{4.10}$$

which correspond to the following topological vacua:

$$\begin{aligned} &\mathcal{M}_{\mathrm{III,a}}[2,k,l,n_f,0] = \\ &\Theta\left(\xi\left(-k-2l-\frac{n_f}{2}\right)\right)U(2)_{k+\frac{n_f}{2},k+\frac{n_f}{2}+2l} \oplus \Theta\left(\xi\left(k+2l-\frac{n_f}{2}\right)\right)U(2)_{k-\frac{n_f}{2},k-\frac{n_f}{2}+2l}. \end{aligned} \tag{4.11}$$

These vacua contribute to the Witten index according to (2.5), namely:

$$\begin{aligned} &\mathbf{I}_{W,\mathrm{III,a}}[2,k,l,n_f,0] = \\ &\Theta\left(\xi\left(-k-2l-\frac{n_f}{2}\right)\right)\mathbf{I}_W\begin{bmatrix} 2 & l & | & k+\frac{n_f}{2} \end{bmatrix} + \Theta\left(\xi\left(k+2l-\frac{n_f}{2}\right)\right)\mathbf{I}_W\begin{bmatrix} 2 & l & | & k-\frac{n_f}{2} \end{bmatrix}, \end{aligned} \tag{4.12}$$

where we used the notation (2.11). We may also have Type III,b vacua with $\sigma_1 > 0$ and $\sigma_2 < 0$, in which case the vacuum equations (4.4) reduce to:

$$\begin{aligned} F(\sigma_1) &= \xi + \left(k + \frac{n_f}{2}\right)\sigma_1 + l(\sigma_1 + \sigma_2) = 0, \\ F(\sigma_1) &= \xi + \left(k - \frac{n_f}{2}\right)\sigma_1 + l(\sigma_1 + \sigma_2) = 0, \end{aligned} \tag{4.13}$$

which have a unique solution:

$$\sigma_1 = -\xi \frac{k - \frac{n_f}{2}}{\left(k+\frac{n_f}{2}\right)\left(k-\frac{n_f}{2}\right) + 2kl} > 0, \qquad \sigma_2 = -\xi \frac{k + \frac{n_f}{2}}{\left(k+\frac{n_f}{2}\right)\left(k-\frac{n_f}{2}\right) + 2kl} < 0. \tag{4.14}$$

The inequalities constraining the appearance of this solution can be simplified to:

$$|k| < \frac{n_f}{2}, \qquad \xi\left(k^2 + 2kl - \frac{1}{4}n_f^2\right) > 0. \tag{4.15}$$

Thus we have the vacua:

$$\begin{aligned} &\mathcal{M}_{\mathrm{III,b}}[2,k,l,n_f,0] = \\ &\Theta\left(\frac{n_f}{2} - |k|\right)\Theta\left(\xi\left(k^2 + 2kl - \frac{1}{4}n_f^2\right)\right)\underbrace{U(1)_{k+l+\frac{n_f}{2}} \times U(1)_{k+l-\frac{n_f}{2}}}_{l}. \end{aligned} \tag{4.16}$$

Using the result (2.10) for abelian CS theories, we have the index contribution:

$$\begin{aligned} &\mathbf{I}_{W,\mathrm{III,b}}[2,k,l,n_f,0] = \\ &\Theta\left(\frac{n_f}{2} - |k|\right)\Theta\left(\xi\left(k^2 + 2kl - \frac{1}{4}n_f^2\right)\right)\mathbf{I}_W\begin{bmatrix} 1 & 0 & | & k+l+\frac{n_f}{2} & l \\ 1 & 0 & | & l & k+l-\frac{n_f}{2} \end{bmatrix}. \end{aligned} \tag{4.17}$$

**Type IV vacua.** Finally, we have to be careful about the 'marginal case' $|k| = \frac{n_f}{2}$, in which case we can have continuous Coulomb branch solutions. Consider first the case $k = \frac{n_f}{2}$. We have a *Type IV,a* solution corresponding to $F_1(\sigma) = F_2(\sigma) = 0$ with $\sigma_1 < 0$, $\sigma_2 < 0$:

$$\sigma_1 + \sigma_2 = -\frac{\xi}{l} < 0, \qquad \text{iff } \xi l > 0. \tag{4.18}$$

Table 1: Witten index for $U(2)_{k,k+2l}$ with $n_f = 4$ fundamentals, for some values of $k, l$. The case with the minimal value $\mathbf{I}_W = 6$ are given in bold. The contributions from Type IV vacua when $\xi > 0$ are shown in red.

| $k\backslash l$ | −10 | −9 | −8 | −7 | −6 | −5 | −4 | −3 | −2 | −1 | 0 | 1 | 2 | 3 | 4 | 5 | 6 | 7 | 8 | 9 | 10 |
|---|---|---|---|---|---|---|---|---|---|---|---|---|---|---|---|---|---|---|---|---|---|
| 0 | 15 | 14 | 13 | 12 | 11 | 10 | 9 | 8 | 7 | **6** | **6** | **6** | 7 | 8 | 9 | 10 | 11 | 12 | 13 | 14 | 15 |
| 1 | 23 | 21 | 19 | 17 | 15 | 13 | 11 | 9 | 7 | **6** | **6** | **6** | 7 | 9 | 11 | 13 | 15 | 17 | 19 | 21 | 23 |
| 2 | 30 | 27 | 24 | 21 | 18 | 15 | 12 | 9 | **6** | **6** | **6** | 6 +3 | 6 +6 | 6 +9 | 6 +12 | 6 +15 | 6 +18 | 6 +21 | 6 +24 | 6 +27 | 6 +30 |
| 3 | 36 | 32 | 28 | 24 | 20 | 16 | 12 | 8 | **6** | **6** | 10 | 14 | 18 | 22 | 26 | 30 | 34 | 38 | 42 | 46 | 50 |
| 4 | 41 | 36 | 31 | 26 | 21 | 16 | 11 | **6** | **6** | 10 | 15 | 20 | 25 | 30 | 35 | 40 | 45 | 50 | 55 | 60 | 65 |
| 5 | 45 | 39 | 33 | 27 | 21 | 15 | 9 | **6** | 10 | 15 | 21 | 27 | 33 | 39 | 45 | 51 | 57 | 63 | 69 | 75 | 81 |
| 6 | 48 | 41 | 34 | 27 | 20 | 13 | **6** | 10 | 14 | 21 | 28 | 35 | 42 | 49 | 56 | 63 | 70 | 77 | 84 | 91 | 98 |
| 7 | 50 | 42 | 34 | 26 | 18 | 10 | 10 | 14 | 20 | 28 | 36 | 44 | 52 | 60 | 68 | 76 | 84 | 92 | 100 | 108 | 116 |
| 8 | 51 | 42 | 33 | 24 | 15 | 10 | 14 | 18 | 27 | 36 | 45 | 54 | 63 | 72 | 81 | 90 | 99 | 108 | 117 | 126 | 135 |
| 9 | 51 | 41 | 31 | 21 | 15 | 14 | 18 | 25 | 35 | 45 | 55 | 65 | 75 | 85 | 95 | 105 | 115 | 125 | 135 | 145 | 155 |
| 10 | 50 | 39 | 28 | 21 | 14 | 18 | 22 | 33 | 44 | 55 | 66 | 77 | 88 | 99 | 110 | 121 | 132 | 143 | 154 | 165 | 176 |

An analogous solution exists in the case $k = -\frac{n_f}{2}$ if $\xi l < 0$. In either case, we find some continuous '$SU(2)$ Coulomb branch' spanned by $\sigma_1 - \sigma_2 \in \mathbb{R}$. This would naively render the Witten index ill-defined. Here, however, we effectively have an $SU(2)_0$ gauge theory at low energy, thus we expect this Coulomb branch to be lifted non-perturbatively [36]. There remains the possibility that some strongly-coupled vacua could survive near $\sigma_1 = \sigma_2 = 0$, and we will thus make a conjecture for their contribution to the Witten index.

In fact, for any fixed CS level $l \in \mathbb{Z}$, the Type IV vacua only arise for one sign of $\xi$. Hence, at any given value of the CS levels, we can simply consider the appropriate sign for $\xi$ in order to compute the Witten index in terms of the vacua of Type I, II and III only. Since the index should be the same for either sign of $\xi$, this allows us to derive the necessary contribution from Type IV vacua:

$$\mathbf{I}_{W,\text{IV}}\left[2, k, l, n_f, 0\right] = \delta_{k, \frac{n_f}{2}} \Theta(\xi l) |l| (n_f - 1) + \delta_{k, -\frac{n_f}{2}} \Theta(-\xi l) |l| (n_f - 1). \tag{4.19}$$

On the other hand, we can only speculate on the nature of the corresponding vacua $\mathcal{M}_{\text{IV}}$ – see also the discussion in section 5.3.

**The full Witten index.** Summing up all the above contributions, the final answer reads:

$$\begin{aligned}\mathbf{I}_W\left[2, k, l, n_f, 0\right] &= \mathbf{I}_{W,\text{I}}\left[2, k, l, n_f, 0\right] + \mathbf{I}_{W,\text{II}}\left[2, k, l, n_f, 0\right] + \mathbf{I}_{W,\text{III,a}}\left[2, k, l, n_f, 0\right] \\ &\quad + \mathbf{I}_{W,\text{III,b}}\left[2, k, l, n_f, 0\right] + \mathbf{I}_{W,\text{IV}}\left[2, k, l, n_f, 0\right].\end{aligned} \tag{4.20}$$

This explicit formula can be compared to the numerical counting of Bethe vacua using Gröbner basis methods [5], and we find perfect agreement. As an example, consider the $U(2)_{k,k+2l}$ theory with 4 fundamentals. Picking $\xi > 0$, the Type I vacua contributes $\chi(\text{Gr}(2,4)) = 6$ and we then have a number of Type II, III, IV vacua, as well as some Type IV vacua for $k = 2$, as indicated in the table 1. This exactly reproduces table 2 of [5].

**Preliminary comments on the phase transition at $\xi = 0$.** While the index is the same for $\xi > 0$ and $\xi < 0$, the structure of the vacuum changes in intricate ways as we change the sign of the FI parameter. As mentioned in the introduction, this is an interesting 3d analogue of the 2d CY/LG correspondence. From our analysis above, we have an explicit form of the full vacuum moduli space $\mathcal{M}$ for each sign of $\xi$ (except when Type IV vacua arise). We show some examples of this in table 2. For instance, looking at the first line with $(k, l) = (0, 10)$, for $\xi > 0$ we have a Higgs and a topological vacuum, which contribute to the Witten index as:

$$\mathbf{I}_W\left[\text{Gr}(2,4)\right] + \mathbf{I}_W\left[U(2)_{-2,18}\right] = 6 + 9 = 15, \tag{4.21}$$

Table 2: Moduli spaces of vacua for $U(2)$ theory coupled with 4 fundamental multiplets and different values of the levels $k$ and $l$. We include both phases of $\xi$, the positive and negative one.

| $k$ | $l$ | $\xi > 0$ phase | $\xi < 0$ phase |
|---|---|---|---|
| 0 | 10 | $\mathrm{Gr}(2,4) \oplus U(2)_{-2,18}$ | $U(2)_{2,22} \oplus \underbrace{U(1)_{12} \times U(1)_8}_{10}$ |
| 1 | 3 | $\mathrm{Gr}(2,4) \oplus \underbrace{U(1)_6 \times U(1)_2}_{3}$ | $U(2)_{3,9}$ |
| 3 | $-2$ | $\mathrm{Gr}(2,4)$ | $\mathbb{CP}^3 \times U(1)_{-1} \oplus U(2)_{5,1}$ |
| 4 | 7 | $\mathrm{Gr}(2,4) \oplus \mathbb{CP}^3 \times U(1)_9 \oplus U(2)_{2,16}$ | $U(2)_{6,20}$ |
| 5 | $-6$ | $\mathrm{Gr}(2,4) \oplus U(2)_{7,-5}$ | $\mathbb{CP}^3 \times U(1)_{-3} \oplus U(2)_{3,-9}$ |
| 6 | $-4$ | $\mathrm{Gr}(2,4)$ | $U(2)_{4,-4}$ |
| 7 | $-9$ | $\mathrm{Gr}(2,4) \oplus U(2)_{9,-9}$ | $\mathbb{CP}^{n_f-1} \times U(1)_{-4} \oplus U(2)_{5,-13}$ |
| 8 | 8 | $\mathrm{Gr}(2,4) \oplus \mathbb{CP}^3 \times U(1)_{14} \oplus U(2)_{6,22}$ | $U(2)_{10,26}$ |
| 9 | 10 | $\mathrm{Gr}(2,4) \oplus \mathbb{CP}^3 \times U(1)_{17} \oplus U(2)_{7,77}$ | $U(2)_{11,31}$ |
| 10 | 5 | $\mathrm{Gr}(2,4) \oplus \mathbb{CP}^3 \times U(1)_{13} \oplus U(2)_{8,18}$ | $U(2)_{12,22}$ |

while the $\xi < 0$ phase consists of two topological vacua:

$$
\mathbf{I}_W \left[ U(2)_{2,22} \right] + \mathbf{I}_W \left[ \underbrace{U(1)_{12} \times U(1)_8}_{10} \right] = 11 + 4 = 15 \, . \tag{4.22}
$$

Note also that, for general values of $(k,l)$, we can have hybrid Higgs-topological vacua for either sign of $\xi$, while the pure Higgs branch only exists for $\xi > 0$.

## 4.2 $U(N_c)_{k,k+lN_c}$ with $n_f$ fundamentals

Let us now discuss the complete solutions to (4.1) in the general case. Following the general discussion above, the **Type I** solution is given by (4.3). Let us now discuss all the other classes of solutions.

**Type II vacua.** Let us take the first $N_c - p$ $\sigma_a$'s to be vanishing, and the last $p$ to be nonzero. In this case, we have the following set of equations (4.1):

$$
\sum_{i=1}^{n_f} Q_a^{\dagger i} Q_i^b = \frac{\delta_a^b}{2\pi} \left( \xi + l \sum_{a'=N_c-p+1}^{N_c} \sigma_{a'} \right), \quad a,b = 1, \cdots, N_c - p \, ,
$$

$$
F_{a'}(\sigma) = \xi + \left( k + \frac{n_f}{2} \right) \sigma_{a'} + l \sum_{b'=N_c-p+1}^{N_c} \sigma_{b'} = 0, \quad a' = N_c - p + 1, \cdots, N_c \, . \tag{4.23}
$$

Let us first assume that all the non-vanishing $\sigma$'s are positive and that $k + \frac{n_f}{2} \neq 0$. In that case, the second line in (4.23) implies that all the non-vanishing $\sigma$'s are equal:

$$
\sigma_{N_c-p+1} = \sigma_{N_c-p+2} = \cdots = \sigma_{N_c} \equiv \sigma^+ \, , \tag{4.24}
$$

so that (4.23) reduces to:

$$\sum_{i=1}^{n_f} Q_a^{\dagger i} Q_i^b = \frac{\delta_a^b}{2\pi}\left(\xi + pl\sigma^+\right), \qquad a, b = 1, \cdots, N_c - p,$$

$$F_+(\sigma) \equiv \xi + \left(k + pl + \frac{n_f}{2}\right)\sigma^+ = 0. \tag{4.25}$$

We then have a single Higgs-topological vacuum with the solution:

$$\sigma^+ = -\frac{\xi}{k + pl + \frac{n_f}{2}} > 0, \qquad \text{iff} \quad \text{sign}(\xi)\left(k + pl + \frac{n_f}{2}\right) < 0 \quad \text{and} \quad k + \frac{n_f}{2} < 0, \tag{4.26}$$

cooresponding to a low-energy effective CS theory $U(p)_{k+\frac{n_f}{2}, k+\frac{n_f}{2}+pl}$ at every point on the Higgs branch $\text{Gr}(N_c - p, n_f)$ that arises from solving the first set of equations in (4.25). The last inequality in (4.26) comes from requiring that the volume of the Grassmannian manifold is positive.

There is also a similar solution where all the the non-vanishing eigenvalues are chosen to be negative (they are then equal as long as $k - \frac{n_f}{2} \neq 0$), with:

$$\sigma^- = -\frac{\xi}{k + pl - \frac{n_f}{2}} < 0, \qquad \text{iff} \quad \text{sign}(\xi)\left(k + pl - \frac{n_f}{2}\right) > 0 \text{ and } k - \frac{n_f}{2} > 0. \tag{4.27}$$

In summary, choosing all possible values of $p$, we have the following Type II,a vacua:

$$\mathcal{M}_{\text{II}}\left[N_c, k, l, n_f, 0\right] =$$

$$\Theta\left(-k - \frac{n_f}{2}\right) \bigoplus_{p=1}^{N_c - 1} \Theta\left(\xi\left(-k - pl - \frac{n_f}{2}\right)\right) \text{Gr}(N_c - p, n_f) \times U(p)_{k+\frac{n_f}{2}, k+\frac{n_f}{2}+pl}$$

$$\oplus \Theta\left(k - \frac{n_f}{2}\right) \bigoplus_{p=1}^{N_c - 1} \Theta\left(\xi\left(k + pl - \frac{n_f}{2}\right)\right) \text{Gr}(N_c - p, n_f) \times U(p)_{k-\frac{n_f}{2}, k-\frac{n_f}{2}+pl}. \tag{4.28}$$

They contribute to the index as:

$$\mathbf{I}_{W,\text{II}}\left[N_c, k, l, n_f, 0\right] =$$

$$\Theta\left(-k - \frac{n_f}{2}\right) \sum_{p=1}^{N_c - 1} \Theta\left(\xi\left(-k - pl - \frac{n_f}{2}\right)\right) \binom{n_f}{N_c - p} \mathbf{I}_W\left[\begin{array}{cc|c} p & l & k + \frac{n_f}{2} \end{array}\right]$$

$$+ \Theta\left(k - \frac{n_f}{2}\right) \sum_{p=1}^{N_c - 1} \Theta\left(\xi\left(k + pl - \frac{n_f}{2}\right)\right) \binom{n_f}{N_c - p} \mathbf{I}_W\left[\begin{array}{cc|c} p & l & k - \frac{n_f}{2} \end{array}\right]. \tag{4.29}$$

As we mentioned at the begining of this section, one should also consider the possibility of having some of the non-vanishing $\sigma$'s to be positive and the other being negative. In this case, we would get Type II,b vacua $\text{Gr}(N_c - p - q, n_f) \times U(p) \times U(q)$. It turns out, however, that in this case the conditions for the TQFTs to appear and for the Grassmannian variety to be of positive size are not mutually compatible, thus there are no such vacua.

**Type III vacua.** Taking all the $\sigma$'s to be non-zero, we obtain various topological vacua. For instance, if all the $\sigma_a$'s are assumed to be positive, then from (4.1) we have:

$$F_a(\sigma) = \xi + \left(k + \frac{n_f}{2}\right)\sigma_a + l\sum_{b=1}^{N_c}\sigma_b = 0, \quad a = 1, \cdots, N_c. \tag{4.30}$$

Any such solution has $\sigma_1 = \cdots = \sigma_{N_c} \equiv \sigma^+$ (assuming $k + \frac{n_f}{2} \neq 0$), and we find:

$$\sigma^+ = -\frac{\xi}{k + lN_c + \frac{n_f}{2}} > 0, \qquad \text{iff } \operatorname{sign}(\xi)\left(k + lN_c + \frac{n_f}{2}\right) < 0. \tag{4.31}$$

Similarly, for $\sigma_a < 0$ (and assuming $k - \frac{n_f}{2} \neq 0$), we obtain a solution:

$$\sigma^- = -\frac{\xi}{k + lN_c - \frac{n_f}{2}}, \qquad \text{iff } \operatorname{sign}(\xi)\left(k + lN_c - \frac{n_f}{2}\right) > 0. \tag{4.32}$$

These two solutions exhausts the Type III,a vacua, which are given by:

$$\begin{aligned}
\mathcal{M}_{\mathrm{III,a}}[N_c, k, l, n_f, 0] &= \Theta\left(\xi\left(-k - lN_c - \frac{n_f}{2}\right)\right) U(N_c)_{k+\frac{n_f}{2}, k+\frac{n_f}{2}+lN_c} \\
&\oplus \Theta\left(\xi\left(k + lN_c - \frac{n_f}{2}\right)\right) U(N_c)_{k-\frac{n_f}{2}, k-\frac{n_f}{2}+lN_c},
\end{aligned} \tag{4.33}$$

which contribute to the Witten index as:

$$\begin{aligned}
\mathbf{I}_{W,\mathrm{III,a}}[N_c, k, l, n_f, 0] &= \Theta\left(\xi\left(-k - lN_c - \frac{n_f}{2}\right)\right) \mathbf{I}_W\left[\begin{array}{cc} N_c & l \mid k + \frac{n_f}{2} \end{array}\right] \\
&+ \Theta\left(\xi\left(k + lN_c - \frac{n_f}{2}\right)\right) \mathbf{I}_W\left[\begin{array}{cc} N_c & l \mid k - \frac{n_f}{2} \end{array}\right].
\end{aligned} \tag{4.34}$$

As for the Type III,b solutions, they are the solutions to the following equations:

$$\begin{aligned}
F_a(\sigma) &= \xi + \left(k + \frac{n_f}{2}\right)\sigma_a + l \sum_{b=1}^{N_c} \sigma_b = 0, \quad a = 1, \cdots, p, \\
F_{a'}(\sigma) &= \xi + \left(k - \frac{n_f}{2}\right)\sigma_{a'} + l \sum_b^{N_c} \sigma_b = 0, \quad a' = p+1, \cdots, N_c,
\end{aligned} \tag{4.35}$$

where we took the first $p$ $\sigma$'s to be of positive sign and the rest to be negative. It again follows that $\sigma_a = \sigma^+$ and $\sigma_{a'} = \sigma^-$ (assuming $|k| \neq \frac{n_f}{2}$), so that (4.35) simplifies to:

$$\begin{aligned}
\xi + \left(k + pl + \frac{n_f}{2}\right)\sigma^+ + (N_c - p)l\sigma^- &= 0, \\
\xi + \left(k + (N_c - p)l - \frac{n_f}{2}\right)\sigma^- + pl\sigma^+ &= 0,
\end{aligned} \tag{4.36}$$

and we have the unique solution:

$$\sigma^+ = -\frac{\xi\left(k - \frac{n_f}{2}\right)}{\mathcal{L}(p, N_c, k, l, n_f)} > 0 \quad \text{and} \quad \sigma^- = -\frac{\xi\left(k + \frac{n_f}{2}\right)}{\mathcal{L}(p, N_c, k, l, n_f)} < 0, \tag{4.37}$$

where we defined the quantity:

$$\mathcal{L}(p, N_c, k, l, n_f) \equiv \left(k + pl + \frac{n_f}{2}\right)\left(k + (N_c - p)l - \frac{n_f}{2}\right) - p(N_c - p)l^2. \tag{4.38}$$

Then, the corresponding Type III,b topological vacua are:

$$\mathcal{M}_{\mathrm{III,b}}[N_c, k, l, n_f, 0] = \Theta\left(\frac{n_f}{2} - |k|\right) \bigoplus_{p=1}^{N_c-1} \Theta\left(\xi\mathcal{L}(p, N_c, k, l, n_f)\right) \underbrace{U(p)_{k+\frac{n_f}{2}, k+\frac{n_f}{2}+pl} \times U(N_c - p)_{k-\frac{n_f}{2}, k-\frac{n_f}{2}+(N_c-p)l}}_{l}, \tag{4.39}$$

Table 3: Witten index for $U(5)_{k,k+5l}$ with $n_f = 7$ fundamentals, for some values of $k, l$. The case with the minimal value $\mathbf{I}_W = 21$ are given in bold. The contributions from Type IV vacua when $\xi > 0$ are shown in red.

| $k \backslash l$ | $-8$ | $-7$ | $-6$ | $-5$ | $-4$ | $-3$ | $-2$ | $-1$ | $0$ | $1$ | $2$ | $3$ | $4$ | $5$ | $6$ | $7$ | $8$ |
|---|---|---|---|---|---|---|---|---|---|---|---|---|---|---|---|---|---|
| $\frac{1}{2}$ | 34 | 32 | 30 | 28 | 26 | 24 | 22 | **21** | **21** | **21** | **21** | 24 | 27 | 30 | 33 | 36 | 39 |
| $\frac{3}{2}$ | 42 | 38 | 34 | 30 | 26 | 23 | 22 | **21** | **21** | **21** | **21** | 23 | 27 | 31 | 35 | 39 | 43 |
| $\frac{5}{2}$ | 55 | 50 | 45 | 40 | 35 | 30 | 25 | **21** | **21** | **21** | **21** | **21** | 26 | 31 | 36 | 41 | 46 |
| $\frac{7}{2}$ | 120 | 105 | 90 | 75 | 60 | 45 | 30 | **21** | **21** | 21 +15 | 21 +30 | 21 +45 | 21 +60 | 21 +75 | 21 +90 | 21 +105 | 21 +120 |
| $\frac{9}{2}$ | 245 | 210 | 175 | 140 | 105 | 70 | 35 | **21** | 56 | 91 | 126 | 161 | 196 | 231 | 266 | 301 | 336 |
| $\frac{11}{2}$ | 455 | 385 | 315 | 245 | 175 | 105 | 35 | 56 | 126 | 196 | 266 | 336 | 406 | 476 | 546 | 616 | 686 |
| $\frac{13}{2}$ | 777 | 651 | 525 | 399 | 273 | 147 | 56 | 126 | 252 | 378 | 504 | 630 | 756 | 882 | 1008 | 1134 | 1260 |
| $\frac{15}{2}$ | 1239 | 1029 | 819 | 609 | 399 | 224 | 91 | 252 | 462 | 672 | 882 | 1092 | 1302 | 1512 | 1722 | 1932 | 2142 |
| $\frac{17}{2}$ | 1869 | 1539 | 1209 | 879 | 584 | 289 | 196 | 462 | 792 | 1122 | 1452 | 1782 | 2112 | 2442 | 2772 | 3102 | 3432 |
| $\frac{19}{2}$ | 2694 | 2199 | 1704 | 1244 | 784 | 324 | 336 | 792 | 1287 | 1782 | 2277 | 2772 | 3267 | 3762 | 4257 | 4752 | 5247 |
| $\frac{21}{2}$ | 3739 | 3024 | 2344 | 1664 | 984 | 409 | 616 | 1287 | 2002 | 2717 | 3432 | 4147 | 4862 | 5577 | 6292 | 7007 | 7722 |

and their contributions to the index read:

$$\mathbf{I}_{W,\mathrm{III,b}}\left[N_c, k, l, n_f, 0\right] =$$

$$\Theta\left(\frac{n_f}{2} - |k|\right) \sum_{p=1}^{N_c-1} \Theta\left(\xi \mathcal{L}(p, N_c, k, l, n_f)\right) \mathbf{I}_W \begin{bmatrix} p & l & \bigm| & k + \frac{n_f}{2} & l \\ N_c - p & l & \bigm| & l & k - \frac{n_f}{2} \end{bmatrix}, \quad (4.40)$$

in terms of the CS index (2.12).

**Type IV vacua.** When $|k| = \frac{n_f}{2}$, semi-classical Coulomb-branch directions open up, rendering our analysis non-reliable. We expect that the actual 'Type IV' vacua are strongly-coupled vacua. As we discussed around (4.19) in the $U(2)$ theory case, we can always calculate the contribution of these (conjectured) vacua to the total Witten index by comparing to the Witten index computed with the opposite sign of FI parameter (which does not have any Type IV contributions). In this way, we find the following contributions, in general:

$$\mathbf{I}_{W,\mathrm{IV}}[N_c, k, l, n_f, 0] = \delta_{k, \frac{n_f}{2}} \Theta(\xi l)|l|\binom{n_f - 1}{N_c - 1} + \delta_{k, -\frac{n_f}{2}} \Theta(-\xi l)|l|\binom{n_f - 1}{N_c - 1}. \quad (4.41)$$

It would be interesting to better understand the physics of these strongly-coupled vacua, but this is left for future work.

**The full Witten index.** Putting all the above contributions together, we have now computed the full Witten index:

$$\mathbf{I}_W\left[N_c, k, l, n_f, 0\right] = \mathbf{I}_{W,\mathrm{I}}\left[N_c, k, l, n_f, 0\right] + \mathbf{I}_{W,\mathrm{II}}\left[N_c, k, l, n_f, 0\right]$$
$$+ \mathbf{I}_{W,\mathrm{III,a}}\left[N_c, k, l, n_f, 0\right] + \mathbf{I}_{W,\mathrm{III,b}}\left[N_c, k, l, n_f, 0\right] + \mathbf{I}_{W,\mathrm{IV}}\left[N_c, k, l, n_f, 0\right]. \tag{4.42}$$

For the reader's convenience, this formula for the index is implemented in a MATHEMATICA [38] notebook attached to this paper. This result can be compared to the Bethe vacua counting, and one finds perfect agreement for all the cases that we checked.[8]

**Example:** $U(5)$ **gauge theory with** $n_f = 7$. Similarly to our discussion of the $U(2)$ theory with 4 fundamentals in subsection 4.1, let us compute the index explicitly in this example – this is shown in table 3. We can also consider the explicit form of the vacua for this theory with

---

[8]Our explicit formula gives us the index for any choice of the parameters, while the Bethe vacua counting is limited, in practice, to relatively low values of the parameters due to the slowless of Gröbner bases algorithms.

Table 4: Moduli spaces of vacua for $U(5)_{k,k+5l}$ with $n_f = 7$ fundamental chiral multiplets, at some values of $k$ and $l$ and for either sign of $\xi$.

| $k$ | $l$ | $\xi > 0$  phase | $\xi < 0$  phase |
|---|---|---|---|
| $\frac{1}{2}$ | 8 | $\text{Gr}(5,7) \oplus \underbrace{U(2)_{4,36} \times U(3)_{-3,21}}_{8}$ | $\underbrace{U(3)_{4,28} \times U(2)_{-3,13}}_{8} \oplus \underbrace{U(4)_{4,36} \times U(1)_5}_{8}$ |
| $\frac{3}{2}$ | 3 | $\text{Gr}(5,7) \oplus \underbrace{U(3)_{5,14} \times U(2)_{-2,4}}_{3}$ | $U(5)_{5,20} \oplus \underbrace{U(4)_{5,17} \times U(1)_1}_{3}$ |
| $\frac{5}{2}$ | 2 | $\text{Gr}(5,7)$ | $U(5)_{6,16} \oplus \underbrace{U(4)_{6,14} \times U(1)_1}_{2}$ |
| $\frac{9}{2}$ | 5 | $\text{Gr}(5,7) \oplus \text{Gr}(4,7) \times U(1)_6$ | $U(5)_{8,33}$ |
| $\frac{11}{2}$ | $-3$ | $\text{Gr}(5,7) \oplus U(5)_{9,-6}$ | $\text{Gr}(4,7) \times U(1)_{-1} \oplus \text{Gr}(3,7) \times U(2)_{2,-4}$ |

either choice of sign for $\xi$, for any given value of the CS levels $k$ and $l$, as shown in table 4. For examples, we see that in the case $(k,l) = (\frac{5}{2}, 2)$, we have a pure Higgs branch in the positive-$\xi$ region and topological vacua on the other side. The index matches on both sides according to:

$$\mathbf{I}_{\text{W}}\left[U(5)_{6,16}\right] + \mathbf{I}_{\text{W}}\left[\underbrace{U(4)_{6,14} \times U(1)_1}_{2}\right] = 16 + 5 = 21 = \chi\left(\text{Gr}(5,7)\right). \tag{4.43}$$

Similar considerations hold for all the other examples shown, and examples of arbitrary complexity can be generated using the attached MATHEMATICA notebook.

# 5   3d $\mathcal{N} = 2$ IR dualities: matching the moduli spaces

Unitary SQCD enjoys an infrared dual description akin to Seiberg duality. The details of the dual theory – also called the 'magnetic' theory, as opposed to the original 'electric' theory – depend crucially on the parameters $k, l$ and $k_c$ [18, 19, 21–23] – see [5] for a detailed discussion. In the previous section, we worked out the vacuum moduli space of SQCD$[N_c, k, l, n_f, 0]$ with vanishing $SU(n_f)$ masses but with non-zero FI parameter. Thus, as an interesting consistency check of our computation, it is natural to ask whether the same result is indeed reproduced in the magnetic theory. Alternatively, the results of this section can be viewed as new detailed checks of the recently proposed dualities for $l \neq 0$ [5, 18, 19]. Therefore, in this section, we study the vacua of the Seiberg-like dual to $U(N_c)_{k,k+lN_c}$ with $n_f$ fundamentals and $\xi \neq 0$. Note that $k_c = \frac{n_f}{2} > 0$, and we then have the dual rank

$$N_c^D \equiv \begin{cases} |k| + \frac{n_f}{2} - N_c\,, & \text{if} \quad |k| \geq \frac{n_f}{2}\,, \\ n_f - N_c\,, & \text{if} \quad |k| \leq \frac{n_f}{2}\,. \end{cases} \tag{5.1}$$

Let us use the notation $\epsilon \equiv \text{sign}(k)$. We have three cases to consider in turn:

(i) **Minimally-chiral case, for $|k| > \frac{n_f}{2}$.** We then have a magnetic theory with gauge group:

$$\underbrace{U(N_c^D)_{-k,-k+\epsilon N_c^D} \times U(1)_{l+\epsilon}}_{\epsilon}\,, \tag{5.2}$$

and with $n_f$ antifundamental chiral multiplets $\widetilde{q}^i$ coupled to the $U(N_c^D)$ factor. The $U(1)$ and $U(N_c^D)$ factors are coupled together by a mixed CS term at level $\epsilon = \pm 1$.

(ii) **Maximally-chiral case, for $|k| < \frac{n_f}{2}$.** In this case, the magnetic theory is simply:

$$U(N_c^D)_{-k, -k+lN_c^D},\qquad(5.3)$$

coupled to the same $n_f$ antifundamental matter fields.

(ii) **Marginally-chiral case, for $|k| = \frac{n_f}{2}$.** In this last case, we have the dual gauge group:

$$\underbrace{U(N_c^D)_{-k, -k+\frac{1}{2}\epsilon N_c^D} \times U(1)_{l+\frac{1}{2}\epsilon}}_{\frac{1}{2}\epsilon},\qquad(5.4)$$

and the same $n_f$ antifundamental chiral multiplets, but we also have an additional chiral multiplet charged under both $U(1) \subset U(N_c^D)$ and the second $U(1)$.

For $l = 0$, these dualities can be simplified further and one recovers the well-known cases studied in [23]. For $|k| > \frac{n_f}{2}$ and $l + \epsilon = 0$, we can integrate out the $U(1)_0$ gauge field in (5.2) and we essentially obtain a $SU(N_c^D)_{-k}$ theory with $n_f$ antifundamentals – this interesting special case will have to be considered separately.

## 5.1 The minimally-chiral dual theory – $|k| > \frac{n_f}{2}$

In the case $|k| > \frac{n_f}{2}$, we have the Nii-Amariti-Rota duality [18,19]:

$$U(N_c)_{k, k+lN_c} \oplus n_f \,\square \quad \longleftrightarrow \quad \underbrace{U(N_c^D)_{-k, -k+\epsilon N_c^D} \times U(1)_{l+\epsilon}}_{\epsilon} \oplus n_f \,\overline{\square}_0.\qquad(5.5)$$

To study the matching of the moduli spaces and of the Witten index across this duality, we first need to analyse the semi-classical vacuum equations for the magnetic theory in (5.5). To write down these equations, we need to recall that the FI parameter of the electric theory is mapped to the FI parameter of the $U(1)_{l+\epsilon}$ theory. (There is no independent topological symmetry for $U(N_c^D)$ because the mixed CS level is $\epsilon = 1$ or $-1$.) We then have:

$$
\begin{aligned}
&-\sigma_a \widetilde{q}_a^i = 0\,, \qquad i = 1, \cdots, n_f\,, \qquad a = 1, \cdots, N_c^D\,,\\
&-\sum_{i=1}^{n_f} \widetilde{q}_i^{\dagger a} \widetilde{q}_b^i = \frac{\delta^a_{\ b}}{2\pi}\left(-k\sigma_a + \epsilon \sum_{b=1}^{N_c^D} \sigma_b - \frac{n_f}{2}|\sigma_a| + \epsilon\tilde{\sigma}\right),\\
&\xi + \epsilon \sum_{c=1}^{N_c^D} \sigma_c + (l+\epsilon)\tilde{\sigma} = 0\,,
\end{aligned}
\qquad(5.6)
$$

where, as in the electric theory, we are taking all matter fields to be massless. Here, $\sigma_b$ and $\tilde{\sigma}$ are the real scalars associated with $U(N_c^D)$ and with the $U(1)_{l+\epsilon}$ factor, respectively.

### 5.1.1 Dual theory for $l + \epsilon \neq 0$

Let us first assume that $l + \epsilon \neq 0$. Then, we can use the last equation in (5.6) to eliminate $\tilde{\sigma}$ from the computation:

$$\tilde{\sigma} = -\frac{\xi}{l+\epsilon} - \frac{\epsilon}{l+\epsilon}\sum_{b=1}^{N_c^D} \sigma_b\,.\qquad(5.7)$$

Substituting this back into (5.6), we find:

$$-\sigma_a \widetilde{q}_a^i = 0, \qquad i = 1, \cdots, n_f, \quad a = 1, \cdots, N_c^D,$$

$$\sum_{i=1}^{n_f} \widetilde{q}_i^{\dagger a} \widetilde{q}_b^i = \frac{\delta^a{}_b}{2\pi} \frac{\epsilon}{l+\epsilon} F_a(\sigma),$$

$$F_a(\sigma) \equiv \xi + \left( \tilde{k} - l + \text{sign}(\sigma_a) \frac{\tilde{n}_f}{2} \right) \sigma_a - l \sum_{b \neq a}^{N_c^D} \sigma_b,$$

(5.8)

where we conveniently defined the parameters:

$$\tilde{k} \equiv \epsilon k (l+\epsilon) \quad \text{and} \quad \tilde{n}_f \equiv \epsilon n_f (l+\epsilon).$$

(5.9)

Note that, while we apparently eliminated the $U(1)_{l+\epsilon}$ factor from the description, this does not mean that it disappears at low energy at this level of the discussion.

We see that the equations (5.8) are very similar to (4.1), therefore we can follow the same strategy to solve them. We will also use a similar typology (*Type I$^D$, II$^D$, ...*). The way vacua are matched across duality may be quite complicated, as we will see. Of course, we know that Higgs vacua should match to Higgs vacua, topological vacua to topological vacua, and indeed they are, but the infrared duality of the larger SQCD theory descends to a geometric and level/rank duality in each vacuum, of the general form:

$$\text{Gr}(p, n_f) \times \{\text{TQFT}\} \quad \longleftrightarrow \quad \text{Gr}(n_f - p, n_f) \times \{\text{level/rank-dual TQFT}\}.$$

(5.10)

The TQFTs on each side are matched precisely through the 3d $\mathcal{N} = 2$ level/rank dualities discussed in section 2.

**Type I$^D$.** This is the vacuum with $\sigma_a = 0$, $\forall a$. We are left with the $D$-term equation:

$$\sum_{i=1}^{n_f} \widetilde{q}_i^{\dagger a} \widetilde{q}_b^i = \frac{\delta^a{}_b}{2\pi} \frac{\xi}{1+\epsilon l},$$

(5.11)

which gives us the Grassmannian $\text{Gr}(N_c^D, n_f)$ if the effective FI term is positive. In addition, (5.7) implies that $\tilde{\sigma}$ obtains a non-zero value, and we then have a $U(1)_{l+\epsilon}$ that survives at low energy. Hence we have a Higgs-topological vacuum:

$$\mathcal{M}_{I^D}^{\min}[N_c, k, l, n_f, 0] = \Theta(\xi(1+\epsilon l)) \, \text{Gr}(N_c^D, n_f) \times U(1)_{l+\epsilon},$$

(5.12)

which contributes to the Witten index as:

$$\mathbf{I}_{W,I}^{\min}[N_c, k, l, n_f, 0] = \Theta(\xi(1+\epsilon l)) \, |l+\epsilon| \binom{n_f}{N_c^D}.$$

(5.13)

**Type II$^D$.** Consider taking $1 \leq p \leq N_c^D - 1$ of the $\sigma$'s to be positive while the remaining ones vanish. Then (5.8) implies that all the non-zero eigenvalues are equal, and the equations simplify to:

$$-\sigma^+ \widetilde{q}_a^i = 0, \qquad i = 1, \cdots, n_f, \qquad a = 1, \cdots, p,$$

$$\sum_{i=1}^{n_f} \widetilde{q}_i^{\dagger b} \widetilde{q}_b^i = \frac{1}{1+\epsilon l} (\xi - pl\sigma^+) > 0, \qquad b, = 1, \cdots, N_c^D - p,$$

$$\xi + \left( \tilde{k} - pl + \frac{\tilde{n}_f}{2} \right) \sigma^+ = 0.$$

(5.14)

A hybrid Higgs-topological solution exists in the window:

$$\text{sign}(\xi)\left(pl - \tilde{k} - \frac{\tilde{n}_f}{2}\right) > 0, \qquad \text{and} \qquad k + \frac{n_f}{2} < 0, \tag{5.15}$$

where the first condition follows from $\sigma^+ > 0$ and the second follows from the positivity of the size of the Grassmannian Higgs branch. A similar solution exists where we take the $p$ non-vanishing $\sigma$'s to be negative, in the window:

$$\text{sign}(\xi)\left(\tilde{k} - pl - \frac{\tilde{n}_f}{2}\right) > 0, \qquad \text{and} \qquad k - \frac{n_f}{2} > 0. \tag{5.16}$$

The $U(1)_{l+\epsilon}$ also survives at low energy, and we then find the branches of hybrid vacua:

$$\mathcal{M}^{\min}_{\text{II}^D}[N_c, k, l, n_f, 0] =$$

$$\Theta\left(-k - \frac{n_f}{2}\right) \bigoplus_{p=1}^{N_c^D-1} \Theta\left(\xi\left(-\tilde{k} + pl - \frac{\tilde{n}_f}{2}\right)\right) \text{Gr}(N_c^D - p, n_f) \times \underbrace{U(p)_{-k-\frac{n_f}{2}, -k-\frac{n_f}{2}+\epsilon p}}_{\epsilon} \times U(1)_{l+\epsilon}$$

$$\oplus \Theta\left(k - \frac{n_f}{2}\right) \bigoplus_{p=1}^{N_c^D-1} \Theta\left(\xi\left(\tilde{k} - pl - \frac{\tilde{n}_f}{2}\right)\right) \text{Gr}(N_c^D - p, n_f) \times \underbrace{U(p)_{-k+\frac{n_f}{2}, -k+\frac{n_f}{2}+\epsilon p}}_{\epsilon} \times U(1)_{l+\epsilon}. \tag{5.17}$$

We see that they contribute to the index as:

$$\mathbf{I}^{\min}_{W, \text{II}^D}[N_c, k, l, n_f, 0] =$$

$$\Theta\left(-k - \frac{n_f}{2}\right) \sum_{p=1}^{N_c^D-1} \Theta\left(\xi\left(-\tilde{k} + pl - \frac{\tilde{n}_f}{2}\right)\right) \binom{n_f}{N_c^D - p} \mathbf{I}_W\begin{bmatrix} p & \epsilon \\ 1 & 0 \end{bmatrix}\begin{matrix} -k-\frac{n_f}{2} & \epsilon \\ \epsilon & l+\epsilon \end{matrix}\end{bmatrix} \tag{5.18}$$

$$+ \Theta\left(k - \frac{n_f}{2}\right) \sum_{p=1}^{N_c^D-1} \Theta\left(\xi\left(\tilde{k} - pl - \frac{\tilde{n}_f}{2}\right)\right) \binom{n_f}{N_c^D - p} \mathbf{I}_W\begin{bmatrix} p & \epsilon \\ 1 & 0 \end{bmatrix}\begin{matrix} -k+\frac{n_f}{2} & \epsilon \\ \epsilon & l+\epsilon \end{matrix}\end{bmatrix},$$

using (2.12) again for the TQFT contributions.

**Type III$^D$.** The third type of solutions correspond to topological vacua. In the analysis of the electric theory, we had solutions of Type III,a and III,b. In the present case, it turns out that only the analogue of Type III,a exists, where all the eigenvalues $\sigma_a$ are taken to have the same sign. In the case where they are all positive, we have $\sigma_a = \sigma^+ > 0$ that solves:

$$\xi + \left(\tilde{k} - lN_c^D + \frac{\tilde{n}_f}{2}\right)\sigma^+ = 0, \qquad \text{sign}(\xi)\left(\tilde{k} - lN_c^D + \frac{\tilde{n}_f}{2}\right) < 0, \tag{5.19}$$

which exists only in the window indicated. Similarly, there is a solution $\sigma_a = \sigma^- < 0$. In total, we find the vacua:

$$\mathcal{M}^{\min}_{\text{III}^D}[N_c, k, l, n_f, 0] =$$

$$\Theta\left(\xi\left(-\tilde{k} + lN_c^D - \frac{\tilde{n}_f}{2}\right)\right) \underbrace{U(N_c^D)_{-k-\frac{n_f}{2}, -k-\frac{n_f}{2}+\epsilon N_c^D}}_{\epsilon} \times U(1)_{l+\epsilon}$$

$$\oplus \Theta\left(\xi\left(\tilde{k} - lN_c^D - \frac{\tilde{n}_f}{2}\right)\right) \underbrace{U(N_c^D)_{-k+\frac{n_f}{2}, -k+\frac{n_f}{2}+\epsilon N_c^D}}_{\epsilon} \times U(1)_{l+\epsilon}, \tag{5.20}$$

Table 5: Matching moduli spaces of vacua across the minimally-chiral duality with $\xi > 0$.

| $N_c$ | $k$ | $l$ | $n_f$ | Electric Side | Magnetic Side |
|---|---|---|---|---|---|
| 3 | 4 | 10 | 6 | $\mathrm{Gr}(3,6) \oplus \mathrm{Gr}(2,6) \times U(1)_{11}$ | $\mathrm{Gr}(3,6) \times \underbrace{U(1)_0 \times U(1)_{11}}_{1} \oplus \mathrm{Gr}(4,6) \times U(1)_{11}$ |
| 5 | $\frac{9}{2}$ | 0 | 7 | $\mathrm{Gr}(5,7) \oplus \mathrm{Gr}(4,6) \times U(1)_1$ | $\mathrm{Gr}(2,7) \times \underbrace{U(1)_0 \times U(1)_1}_{1} \oplus \mathrm{Gr}(3,7) \times U(1)_1$ |
| 5 | $-\frac{9}{2}$ | 2 | 7 | $\mathrm{Gr}(5,7) \oplus U(5)_{-8,5}$ | $\mathrm{Gr}(2,7) \oplus \underbrace{U(1)_0 \times U(1)_1}_{-1} \oplus \underbrace{U(3)_{8,5} \times U(1)_1}_{-1}$ |
| 6 | 5 | $-4$ | 8 | $\mathrm{Gr}(6,8) \oplus U(6)_{9,-15}$ | $\mathrm{Gr}(2,8) \times \underbrace{U(1)_0 \times U(1)_{-3}}_{1} \oplus \underbrace{U(3)_{-9,-6} \times U(1)_{-3}}_{1}$ |

Table 6: Matching moduli spaces of vacua across the minimally-chiral duality with $\xi < 0$.

| $N_c$ | $k$ | $l$ | $n_f$ | Electric Side | Magnetic Side |
|---|---|---|---|---|---|
| 3 | 4 | 10 | 6 | $U(3)_{7,37}$ | $\underbrace{U(4)_{-7,-3} \times U(1)_{11}}_{1}$ |
| 5 | $\frac{9}{2}$ | 0 | 7 | $U(5)_{8,8}$ | $\underbrace{U(3)_{-8,-5} \times U(1)_1}_{1}$ |
| 5 | $-\frac{9}{2}$ | 2 | 7 | $\mathrm{Gr}(4,7) \oplus U(1)_1$ | $\mathrm{Gr}(3,7) \oplus U(1)_1$ |
| 6 | 5 | $-4$ | 8 | $\mathrm{Gr}(5,8) \times U(1)_{-3}$ | $\mathrm{Gr}(3,8) \times U(1)_{-3}$ |

which contribute to the index as:

$$
\begin{aligned}
\mathbf{I}^{\min}_{W,\mathrm{III}^D}[N_c,k,l,n_f,0] &= \Theta\left(\xi\left(-\tilde{k}+lN_c^D - \frac{\tilde{n}_f}{2}\right)\right)\mathbf{I}_W\left[\begin{array}{cc|cc} N_c^D & \epsilon & -k-\frac{n_f}{2} & \epsilon \\ 1 & 0 & \epsilon & l+\epsilon \end{array}\right] \\
&\oplus \Theta\left(\xi\left(\tilde{k}-lN_c^D - \frac{\tilde{n}_f}{2}\right)\right)\mathbf{I}_W\left[\begin{array}{cc|cc} N_c^D & \epsilon & -k+\frac{n_f}{2} & \epsilon \\ 1 & 0 & \epsilon & l+\epsilon \end{array}\right].
\end{aligned}
\tag{5.21}
$$

**Full Witten index.** One can check that there are no more solutions, so that the full Witten index of the minimally-chiral dual theory takes the form:

$$
\begin{aligned}
\mathbf{I}^{\min}_W[N_c,k,l,n_f,0] = \mathbf{I}^{\min}_{W,\mathrm{I}^D}[N_c,k,l,n_f,0] &+ \mathbf{I}^{\min}_{W,\mathrm{II}^D}[N_c,k,l,n_f,0] \\
&+ \mathbf{I}^{\min}_{W,\mathrm{III}^D}[N_c,k,l,n_f,0].
\end{aligned}
\tag{5.22}
$$

**Matching across the duality.** Using these results, one can check that the moduli spaces match exactly between the electric and magnetic description, for either sign of the FI parameter, as predicted by the duality 5.5. For instance, in the case $\xi > 0$, the electric theory vacua always include the pure Higgs branch 4.3. The dual description is simply through the Grassmannian duality:

$$
\mathrm{Gr}(N_c,n_f) \qquad \longleftrightarrow \qquad \mathrm{Gr}(n_f-N_c,n_f).
\tag{5.23}
$$

The left-hand-side Higgs branch arises as a Type $\mathrm{II}^D$ vacuum with a TQFT that happens to be trivial (with Witten index $\mathbf{I}_W = 1$), namely the vacua in (5.17) with $p = |k| - \frac{n_f}{2}$.

Here, let us simply display this general matching in some examples for $\xi > 0$ (table 5) and $\xi < 0$ (table 6) . More examples are listed in appendix B, and more can be generated using the MATHEMATICA notebook attached. For instance, looking at the last case, $(N_c, k, l, n_f) = (6, 5, -4, 8)$, in table 5, we see that the matching of the vacua follows from the dualities:

$$
\begin{aligned}
\mathrm{Gr}(6,8) &\longleftrightarrow \mathrm{Gr}(2,8) \times \underbrace{U(1)_0 \times U(1)_{-3}}_{1}\,, \\
U(6)_{9,-15} &\longleftrightarrow \underbrace{U(3)_{-9,-6} \times U(1)_{-3}}_{1}\,.
\end{aligned}
\tag{5.24}
$$

Here, the TQFT on the right-hand-side of the first line has a single state, and on the second line we have an instance of the level/rank duality (2.13) (here for $(N, k, l) = (6, 9, -4)$). Similar matching holds for every component of the moduli space for every theory.

### 5.1.2 Dual theory for $l + \epsilon = 0$

We must treat separately the case $|k| > \frac{n_f}{2}$ with $l + \epsilon = 0$. In this case, we have a $U(1)_0$ coupled to the trace of the $U(N_c^D)$ vector multiplet by a BF term, and integrating it out imposes:

$$
\mathrm{tr}\left(A^{U(N_c^D)}\right) = -\epsilon A_T\,, \qquad \sum_{b=1}^{N_c^D} \sigma_b = -\epsilon \xi\,.
\tag{5.25}
$$

This gets rid of the $U(1) \subset U(N_c^D)$, and the duality (5.5) becomes:

$$
U(N_c)_{k, k-\epsilon N_c} \oplus n_f \,\square \longleftrightarrow SU(N_c^D)_{-k} \oplus n_f \,\overline{\square}.
\tag{5.26}
$$

In this formulation, the topological symmetry of the electric theory maps to the baryonic symmetry of the $SU(N_c^D)$ magnetic dual theory. (A closely related non-supersymmetric duality was first discussed in [39].)

This magnetic theory has the same moduli space of vacua as some $SU(n_c)_k$ theory coupled to $n_f$ fundamental chiral multiplets (this is just a CP transformation):

$$
SU(n_c)_k \oplus n_f \,\square\,.
\tag{5.27}
$$

We leave a more detailed analysis of this theory for the motivated reader. Here, we simply conjecture an explicit formula for its flavoured Witten index:

$$
\mathbf{I}_W\left[SU(n_c)_k \oplus n_f \,\square\right] = \binom{|k| + \frac{n_f}{2} - 1}{n_c - 1}\,, \qquad \text{if } |k| > \frac{n_f}{2}\,.
\tag{5.28}
$$

Of course, this reduces to the known result (2.4) for the pure CS theory if $n_f = 0$. We checked that, for $n_c = N_c^D = |k| + \frac{n_f}{2} - N_c$, this indeed matches with the index computed by the complicated formula (4.42) in the electric theory. For $n_c = 2$, the formula (5.28) was derived in [3].

## 5.2 The maximally-chiral dual theory – $|k| < \frac{n_f}{2}$

In the maximally-chiral case with $k < \frac{n_f}{2}$, we have the duality [5]:

$$
U(N_c)_{k, k+lN_c} \oplus n_f \,\square \longleftrightarrow U(N_c^D)_{-k, -k+lN_c^D} \oplus n_f \,\overline{\square}\,,
\tag{5.29}
$$

with $N_c^D \equiv n_f - N_c$. The semi-classical equations for the magnetic theory take the form:

$$-\sigma_a \widetilde{q}_a^i = 0, \qquad i = 1, \cdots, n_f, \qquad a = 1, \cdots, N_c^D$$

$$-\sum_{i=1}^{n_f} \widetilde{q}_i^{\dagger a} \widetilde{q}_b^i = \frac{\delta^a_{\ b}}{2\pi} \left( -\xi - k\sigma_a - \frac{n_f}{2}|\sigma_a| + l \sum_{c=1}^{N_c^D} \sigma_c \right). \tag{5.30}$$

The sign in front of the FI term is due to the fact that the topological current flips sign across this duality [5]. We then see that the analysis of the solutions will be completely similar to the one for the electric theory, after taking into account that we effectively changed the sign of $l$ relative to $k$, and that we have to integrate out matter fields according to the effective real mass $-\sigma_a$. Here we denote the types of vacua by $I_D$, $II_D$,..., not to be confused with the minimally-chiral case discussed above.

**Type $I_D$ vacua: The Higgs branch.** Taking all the $\sigma$'s to vanish, we have the usual Higgs branch equation that gives us the Grassmannian:

$$\mathcal{M}_{I_D}^{\max} = \Theta(\xi) \operatorname{Gr}\left(N_c^D, n_f\right), \qquad\qquad \mathbf{I}_{W,I_D}^{\max} = \Theta(\xi) \binom{n_f}{N_c^D}. \tag{5.31}$$

Note that this exactly matches the Higgs branch vacuum (4.3) in the electric theory, due to the Grassmannian duality $\operatorname{Gr}(N_c, n_f) \cong \operatorname{Gr}(n_f - N_c, n_f)$.

**Type $II_D$ vacua.** There are no Type $II_D$ vacua in this magnetic theory because of the constraint $|k| \neq \frac{n_f}{2}$, as we can see already from the computation in (4.28).

**Type $III_D$ vacua.** As in the electric theory, we have the Type $III,a_D$ solutions when all the $\sigma$'s are of the same sign. They give us:

$$\mathcal{M}_{III,a_D}^{\max}[N_c, k, l, n_f, 0] = \Theta\left(\xi\left(-k + N_c^D l - \frac{n_f}{2}\right)\right) U(N_c^D)_{-k-\frac{n_f}{2}, -k-\frac{n_f}{2}+N_c^D l}$$

$$\oplus \Theta\left(\xi\left(k - N_c^D l - \frac{n_f}{2}\right)\right) U(N_c^D)_{-k+\frac{n_f}{2}, -k+\frac{n_f}{2}+N_c^D l}, \tag{5.32}$$

and contribute to the index as:

$$\mathbf{I}_{W,III,a_D}^{\max}[N_c, k, l, n_f, 0] = \Theta\left(\xi\left(-k + N_c^D l - \frac{n_f}{2}\right)\right) \mathbf{I}_W\left[\begin{array}{cc|c} N_c^D & l & -k - \frac{n_f}{2} \end{array}\right]$$

$$+ \Theta\left(\xi\left(k - N_c^D l - \frac{n_f}{2}\right)\right) \mathbf{I}_W\left[\begin{array}{cc|c} N_c^D & l & -k + \frac{n_f}{2} \end{array}\right]. \tag{5.33}$$

Finally, we can also have Type $III,b_D$ solutions that give us the topological vacua:

$$\mathcal{M}_{III,b_D}^{\max}[N_c, k, l, n_f, 0] =$$

$$\bigoplus_{p=1}^{N_c^D-1} \Theta\left(\xi \mathcal{L}(p, N_c^D, k, -l, n_f)\right) U(p)_{-k-\frac{n_f}{2}, -k-\frac{n_f}{2}+lp} \times U(N_c^D - p)_{\underbrace{-k+\frac{n_f}{2}, -k+\frac{n_f}{2}+l(N_c^D-p)}_{l}}, \tag{5.34}$$

with the quadratic function $\mathcal{L}$ defined in (4.38), which contribute to the index as:

$$\mathbf{I}_{W,III,b_D}^{\max}[N_c, k, l, n_f, 0] =$$

$$\sum_{p=1}^{N_c^D-1} \Theta\left(\xi \mathcal{L}(p, N_c^D, k, -l, n_f)\right) \mathbf{I}_W\left[\begin{array}{cc|cc} p & l & -k-\frac{n_f}{2} & l \\ N_c^D - p & l & l & -k+\frac{n_f}{2} \end{array}\right]. \tag{5.35}$$

Table 7: Matching moduli spaces of vacua across the maximally-chiral duality with $\xi > 0$.

| $N_c$ | $k$ | $l$ | $n_f$ | Electric Side | Magnetic Side |
|---|---|---|---|---|---|
| 3 | 0 | 10 | 6 | $\mathrm{Gr}(3,6) \oplus U(3)_{-3,27}$ $\oplus \underbrace{U(1)_{13} \times U(2)_{-3,17}}_{10}$ | $\mathrm{Gr}(3,6) \oplus U(3)_{-3,27}$ $\oplus \underbrace{U(1)_{13} \times U(2)_{-3,17}}_{10}$ |
| 3 | 1 | 3 | 6 | $\mathrm{Gr}(3,6) \oplus \underbrace{U(1)_7 \times U(2)_{-2,4}}_{3}$ | $\mathrm{Gr}(3,6) \oplus U(3)_{-4,5}$ |
| 3 | 2 | 1 | 6 | $\mathrm{Gr}(3,6)$ | $\mathrm{Gr}(3,6)$ |
| 5 | $\frac{5}{2}$ | 0 | 7 | $\mathrm{Gr}(5,7)$ | $\mathrm{Gr}(2,7)$ |
| 5 | $-\frac{3}{2}$ | 2 | 7 | $\mathrm{Gr}(5,7) \oplus U(5)_{-5,5}$ | $\mathrm{Gr}(2,7) \oplus U(2)_{-2,2}$ |
| 6 | 2 | -4 | 8 | $\mathrm{Gr}(6,8) \oplus U(6)_{6,-18}$ $\oplus \underbrace{U(5)_{6,-14} \times U(1)_{-6}}_{-4}$ | $\mathrm{Gr}(2,8) \oplus U(2)_{2,-6}$ $\oplus \underbrace{U(1)_{-10} \times U(1)_{-2}}_{-4}$ |

Table 8: Matching moduli spaces of vacua across the maximally-chiral duality with $\xi < 0$.

| $N_c$ | $k$ | $l$ | $n_f$ | Electric Side | Magnetic Side |
|---|---|---|---|---|---|
| 3 | 0 | 10 | 6 | $U(3)_{3,33} \oplus \underbrace{U(1)_7 \times U(2)_{3,23}}_{10}$ | $U(3)_{3,33} \oplus \underbrace{U(1)_7 \times U(2)_{3,23}}_{10}$ |
| 3 | 1 | 3 | 6 | $U(3)_{4,13} \oplus \underbrace{U(2)_{4,10} \times U(1)_1}_{3}$ | $\underbrace{U(1)_{-1} \times U(2)_{2,8}}_{3} \oplus \underbrace{U(2)_{-4,2} \times U(1)_5}_{3}$ |
| 3 | 2 | 1 | 6 | $\underbrace{U(2)_{5,7} \times U(1)_0}_{1} \oplus U(3)_{5,8}$ | $U(3)_{-5,-2} \oplus \underbrace{U(2)_{-5,-3} \times U(1)_2}_{1}$ |
| 5 | $\frac{5}{2}$ | 0 | 7 | $U(5)_{6,6} \oplus U(4)_{6,6} \times U(1)_{-1}$ | $U(1)_{-6} \times U(1)_1 \oplus U(2)_{-6,-6}$ |
| 5 | $-\frac{3}{2}$ | 2 | 7 | $\underbrace{U(1)_4 \times U(4)_{-5,3}}_{2} \oplus \underbrace{U(2)_{2,6} \times U(3)_{-5,1}}_{2}$ | $U(2)_{5,9} \oplus \underbrace{U(1)_0 \times U(1)_7}_{2}$ |
| 6 | 2 | −4 | 8 | $\underbrace{U(4)_{6,-10} \times U(2)_{-2,-10}}_{-4}$ | $U(2)_{-6,-14}$ |

**Full Witten index and matching across the duality.** Adding the above contributions, the Witten index is given by:

$$
\mathbf{I}_W^{\max}[N_c,k,l,n_f,0] = \mathbf{I}_{W,\mathrm{I}_D}^{\max}[N_c,k,l,n_f,0] + \mathbf{I}_{W,\mathrm{III},\mathrm{a}_D}^{\max}[N_c,k,l,n_f,0]
$$
$$
+ \mathbf{I}_{W,\mathrm{III},\mathrm{b}_D}^{\max}[N_c,k,l,n_f,0].
\tag{5.36}
$$

It is not complicated to prove that the vacua match one-to-one across the duality. We display some examples of dual vacua for $\xi > 0$ and $\xi < 0$ in table 7 and 8, respectively; see also appendix B. In particular, note that if $\xi < 0$ we can only have topological vacua, in which case the matching of the vacua follows from the level/rank duality (2.22).

### 5.3 The marginally-chiral dual theory – $|k| = \frac{n_f}{2}$

Finally, let us briefly discuss the case $k = \epsilon \frac{n_f}{2}$. We have the marginally-chiral duality [5, 19]:

$$
\begin{aligned}
U(N_c)_{\epsilon\frac{n_f}{2},\epsilon\frac{n_f}{2}+lN_c} \,\oplus n_f\,\square \\
\longleftrightarrow \quad \underbrace{U(n_f-N_c)_{-\epsilon\frac{n_f}{2},-\epsilon\frac{n_f}{2}+\frac{1}{2}\epsilon(n_f-N_c)} \times U(1)_{l+\frac{1}{2}\epsilon}}_{\frac{1}{2}\epsilon} \,\oplus n_f\,\overline{\square}_0 \oplus \mathbf{det}_{+1}\,.
\end{aligned}
\tag{5.37}
$$

The dual theory involves the 'baryon' $\mathcal{B}$ that transforms in the determinant representation of $U(N_c^D)$ ($N_c^D \equiv n_f - N_c$) and has charge 1 under the additional $U(1)$ factor. Let us choose $k = \frac{n_f}{2} > 0$ for definiteness. The semiclassical vacuum equations then read:

$$
-\sigma_a \widetilde{q}_a^i = 0\,, \qquad i = 1,\cdots,n_f\,, \qquad a = 1,\cdots,N_c^D\,,
$$

$$
\left(\sum_{a=1}^{N_c^D} \sigma_a + \tilde{\sigma}\right)\mathcal{B} = 0\,,
$$

$$
\delta^a{}_b\,\mathcal{B}^\dagger \mathcal{B} - \sum_{i=1}^{n_f} \widetilde{q}_i^{\dagger a}\widetilde{q}_b^i = \frac{\delta^a{}_b}{2\pi}\left(-\frac{n_f}{2}\sigma_a + \frac{1}{2}\sum_{c=1}^{N_c^D}\sigma_c + \frac{1}{2}\tilde{\sigma} - \frac{n_f}{2}|\sigma_a| + \frac{1}{2}\left|\sum_{c=1}^{N_c^D}\sigma_c + \tilde{\sigma}\right|\right)\,,
\tag{5.38}
$$

$$
\mathcal{B}^\dagger \mathcal{B} = \frac{1}{2\pi}\left(\xi + \frac{1}{2}\sum_{c=1}^{N_c^D}\sigma_c + \left(l + \frac{1}{2}\right)\tilde{\sigma} + \frac{1}{2}\left|\sum_{c=1}^{N_c^D}\sigma_c + \tilde{\sigma}\right|\right)\,.
$$

We will not attempt to solve these equations here. Instead, let us simply consider a very special case for which these equations trivialise.

#### 5.3.1 Special case $N_c^D = 0$

If we choose $n_f = N_c = 2k$ in the electric theory, the dual theory is simply the abelian theory:

$$
U(1)_{l+\frac{1}{2}} \oplus \mathcal{B}\,,
\tag{5.39}
$$

which was discussed in section 3.2. We have the vacua:

$$
\Theta(\xi)\,\mathbb{CP}^0 \oplus \Theta(-\xi(l+1))\,U(1)_{l+1} \oplus \Theta(\xi l)\,U(1)_l\,,
\tag{5.40}
$$

and the index:

$$
\mathbf{I}_W\left[n_f,\frac{n_f}{2},l,n_f,0\right] = \frac{1}{2} + \left|l + \frac{1}{2}\right| = \begin{cases} l+1\,, & \text{if } l \geq 0\,, \\ |l|\,, & \text{if } l < 0\,. \end{cases}
\tag{5.41}
$$

Let us compare this to the results of section 4.2. One directly sees that the various types of vacua of the electric theory contribute as:

$$
\begin{aligned}
\mathbf{I}_{W,\mathrm{I}}\left[n_f,\frac{n_f}{2},l,n_f,0\right] &= \Theta(\xi)\,, \\
\mathbf{I}_{W,\mathrm{III,a}}\left[n_f,\frac{n_f}{2},l,n_f,0\right] &= \Theta(-\xi(l+1))\,|l+1|\,, \\
\mathbf{I}_{W,\mathrm{IV}}\left[n_f,\frac{n_f}{2},l,n_f,0\right] &= \Theta(\xi l)\,|l|\,,
\end{aligned}
\tag{5.42}
$$

with no contribution from the Type II and III,b vacua. Adding all the contributions as in (4.42), this matches exactly with (5.41). We also see that, for $N_c^D = 0$, the strongly-coupled vacua (Type IV vacua) of the electric theory, whose existence we conjectured in section 4, correspond to an ordinary topological vacua in the dual description – namely the $U(1)_l$ TQFT in (5.40).

### 5.3.2 Case $N_c^D > 0$

For $k = \frac{n_f}{2}$ and $N_c = n_f - N_c > 0$, however, we find that the electric 'Type IV' vacua also reappear as 'quantum vacua' in the magnetic description. In the electric description, as discussed in section 4.2, we have the vacua:

$$\mathcal{M}^{\text{elec}} = \Theta(\xi)\,\text{Gr}(N_c, n_f) \oplus \Theta(-\xi(n_f + lN_c))\,U(N_c)_{n_f, n_f + lN_c} \oplus \Theta(\xi l)\,\mathcal{M}_{\text{IV}}, \qquad (5.43)$$

where $\mathcal{M}_{\text{IV}}$ denotes the 'quantum vacua'. Looking at the magnetic description, it is useful to write the equations (5.38) in terms of the effective real mass for the 'baryon' field $\mathcal{B}$:

$$\sigma_B \equiv \sum_{a=1}^{N_c^D} \sigma_a + \tilde{\sigma}. \qquad (5.44)$$

We can then analyse the theory by moving along the $\sigma_B$ line. The first obvious solution is for $\sigma_B = 0 = \sigma_a$, in which case we have the Higgs vacuum:

$$\Theta(\xi)\,\text{Gr}(N_c^D, n_f), \qquad (5.45)$$

which obviously matches the first term in (5.43). For $\sigma_B > 0$, there is also a single solution with $\sigma_a > 0$ that corresponds to the TQFT:

$$\Theta(-\xi((l+1)n_f - lN_c^D))\,\underbrace{U(N_c^D)_{-n_f, -n_f + N_c^D} \times U(1)_{l+1}}_{1}, \qquad (5.46)$$

which reproduces exactly the second vacuum in (5.43) (after a level/rank duality). Finally, we find that, for any $N_c^D > 0$, there is a continuous Coulomb-branch vacuum for $\sigma_B < 0$ and $\sigma_a < 0$. Assuming $\sigma_B < 0$, we can integrate out $\mathcal{B}$ to obtain a decoupled $U(1)_l$ sector (with the FI parameter $\xi$). This accounts for the factor $\Theta(\xi)|l|$ of the Witten index of the conjectured Type-IV vacuum:

$$\mathbf{I}_{W,\text{IV}} = \Theta(\xi l)\,|l|\binom{n_f - 1}{N_c - 1} = \Theta(\xi l)\,|l|\binom{n_f - 1}{N_c^D}, \qquad (5.47)$$

but we do not have a complete understanding of this vacuum in the magnetic theory.

## Acknowledgements

**Funding information** CC is a Royal Society University Research Fellow and a Birmingham Fellow, and his work is supported by the University Research Fellowship Renewal 2022 'Singularities, supersymmetry and SQFT invariants'. The work of OK is supported by the School of Mathematics at the University of Birmingham.

## A $\mathcal{N} = 0$ level/rank dualities

For completeness, here we present the $\mathcal{N} = 0$ (non-supersymmetric) version of the level/rank dualities between Chern-Simons theories discussed in the main text. They are obtained from the $\mathcal{N} = 2$ dualities by integrating out the gauginos in vector multiplets. Recall that the $\mathcal{N} = 2$ supersymmetric CS interaction includes a term quadratic in the gauginos, which amounts to a real mass:

$$m_\lambda = -\frac{k}{4\pi}. \qquad (A.1)$$

Hence, we have an infrared $\mathcal{N} = 0$ description in which the CS levels are shifted according to the sign of $m_\lambda$. More generally, $m_\lambda$ is a non-trivial mass matrix, which we should then diagonalise. We follow the conventions of [5] for the $\mathcal{N} = 2$ CS contact terms, except that here we denote by the letters $k, l, \cdots$ the $\mathcal{N} = 2$ CS levels (for either the gauge or flavour symmetries), and by the capital letters $K, L, \cdots$ all the $\mathcal{N} = 0$ CS levels.

## A.1 Level/rank dual for $U(N)_{K,K+LN}$

Consider the $U(N)_{k,k+lN}$ $\mathcal{N} = 2$ supersymmetric pure CS theory. Without loss of generality, consider:

$$k > 0, \qquad K \equiv k - N \geq 0. \tag{A.2}$$

The $\mathcal{N} = 2$ version of the level/rank duality is a specialisation of the Nii duality [18],

$$\mathcal{N} = 2 : \qquad U(N)_{k,k+lN} \qquad \longleftrightarrow \qquad \underbrace{U(k-N)_{-k,-N} \times U(1)_{l+1}}_{1}, \tag{A.3}$$

with the non-zero $\mathcal{N} = 2$ CS contact terms derived in [5], which amounts to having $\kappa_{RR}^{(e)} = \frac{1}{2}N^2 = \frac{1}{2}\kappa_g^{(e)}$ on the electric side, and $\kappa_{RR}^{(m)} = -\frac{1}{2}(k-N)^2$ and $\kappa_g^{(m)} = N^2 + 1 - k^2$ on the magnetic side. Integrating out the gaugino in the $U(N)$ theory, we shift the CS levels to $K \equiv k - N$ and $L \equiv l + 1$, and $K_{RR}^{(e)} = K_g^{(e)} = 0$ in the IR. On the magnetic side, integrating out the gauginos similarly shifts the CS level for the $SU(K)$ part of the gauge group, so that we have the level/rank duality:

$$\mathcal{N} = 0 : \qquad U(N)_{K,K+LN} \qquad \longleftrightarrow \qquad \underbrace{U(K)_{-N,-N} \times U(1)_L}_{1}, \tag{A.4}$$

which was first described in [40]. By a straightforward computation, we can check that the $U(1)_R$ symmetry decouples entirely, and we find the relative CS level $K_g = -2kN$. Similarly to [27], we can write this as:

$$\frac{1}{2\pi}A_T \wedge \text{Tr}(F_{U(N)}) \qquad \longleftrightarrow \qquad \frac{1}{2\pi}A_T \wedge F_{U(1)} - 2kN \, \text{CS}_{\text{grav}}, \tag{A.5}$$

where we only wrote down terms that depend on background fields. Here the background gauge field $A_T$ couples to the topological symmetry current on either side of the duality, which is $\text{Tr}(F_{U(N)})$ on the electric side, and $F_{U(1)}$ the field strength for the $U(1)$ factor in $U(K) \times U(1)$ on the magnetic side.[9]

For $L = \pm 1$, we can use the fact that $U(1)_{\pm 1}$ is an 'almost trivial' theory [27,41] to simplify (A.4), and we arrive at the following dualities:

$$\mathcal{N} = 0 : \qquad U(N)_{K,K\pm N} \qquad \longleftrightarrow \qquad U(K)_{-N,-N\mp K}, \tag{A.7}$$

with:

$$\frac{1}{2\pi}A_T \wedge \text{Tr}(F_{U(N)}) \qquad \longleftrightarrow \qquad \mp\frac{1}{2\pi}A_T \wedge \text{Tr}(F_{U(K)}) \mp \frac{1}{4\pi}A_T \wedge dA_T - (2kN \pm 2)\text{CS}_{\text{grav}}. \tag{A.8}$$

---

[9]To derive this result starting from the $\mathcal{N} = 2$ version of the duality, one must take into account the fact that the abelian part of the magnetic gauge group is $U(1) \times U(1)$ with a non-trivial mass matrix

$$M_\lambda = \begin{pmatrix} KN & -K \\ -K & -L \end{pmatrix} \tag{A.6}$$

for the abelian gauginos. We use the fact that the two eigenvalues $m_\pm$ are such that $m_+ > 0$ and $m_- < 0$ if $K + LN > 0$, while $m_\pm > 0$ if $K + LN < 0$.

For $L = 1$, this is a special case of the Giveon-Kutasov duality (the $\mathcal{N} = 2$ duality $U(N)_k \longleftrightarrow U(k-N)_{-k}$) reduced to $\mathcal{N} = 0$, while for $L = -1$ this duality was first derived in [27].

Finally, for $L = 0$, we have $U(1)_0$ theory coupled by a BF term to the $U(K)$ factor in (A.4), and the path integral over that gauge field imposes the constraint [41]:

$$\text{Tr}\left(A^{U(K)}\right) = -A_T, \tag{A.9}$$

on the $U(K)$ gauge field, which gives us to the standard level/rank duality:

$$\mathcal{N} = 0 : \qquad U(N)_{K,K} \qquad \longleftrightarrow \qquad SU(K)_{-N}. \tag{A.10}$$

More precisely, we truly have an $SU(K)$ theory only when the background field $A_T$ is set to zero, while having $A_T$ generic is important to keep track of the $U(1)_T$ symmetry across the duality [27].

## A.2 Level/rank duality for $U(N) \times U(N')$

Let us also discuss the $\mathcal{N} = 0$ version of the duality for the $U(N) \times U(N')$ theory derived in section 2.2. First of all, we have the $\mathcal{N} = 0$ abelian duality:

$$\mathcal{N} = 0 : \qquad \underbrace{U(1)_{l+1} \times U(1)_{l-1}}_{l} \qquad \longleftrightarrow \qquad \begin{cases} K_{TT} = l - 1, & K_{T'T'} = l + 1, \\ K_{TT'} = -l, & K_g = 2 + c_g, \end{cases} \tag{A.11}$$

where the magnetic theory is an empty theory with the contact terms as indicated for the background gauge fields $A_T$ and $A'_T$ coupling to the topological symmetry currents $F$ and $F'$. Here $c_g \in \mathbb{Z}$ is a constant that we did not determine, although we know that $c_g = 0$ when $l = 0$. The $\mathcal{N} = 0$ duality (A.11) directly follows from (2.21), taking into account the fact that the gaugino mass matrix has eigenvalues $m_\pm = -l \pm \sqrt{l^2 + 1}$. (Requiring that the $U(1)_R$ symmetry consistenly decouples then fixes $K_{RR}$ in (2.21).)

Finally, we consider the $U(N)_{k,k+lN} \times U(N')_{k',k'+lN'}$ $\mathcal{N} = 2$ theory with mixed CS level $l$. Let us fix $k > 0$ and $k' < 0$, without loss of generality. Integrating out the gauginos, we obtain the $\mathcal{N} = 0$ levels:

$$\underbrace{U(N)_{K,K+(l+1)N} \times U(N')_{-K',-K'+(l-1)N'}}_{l}, \tag{A.12}$$

where we choose $K \geq 0$ and $K' \geq 0$ for simplicity of notation. Then, the $\mathcal{N} = 2$ duality (2.22) gives us the generalised level/rank duality:

$$\mathcal{N} = 0 : \quad \underbrace{U(N)_{K,K+(l+1)N} \times U(N')_{-K',-K'+(l-1)N'}}_{l}$$
$$\longleftrightarrow \quad \underbrace{U(K)_{-N,-N+(l-1)K} \times U(K')_{N',N'+(l+1)K'}}_{l}. \tag{A.13}$$

Of course, this can also be derived directly in the $\mathcal{N} = 0$ context, starting from (A.4) and following the same logic as in section 2.2.

# B  Matching moduli spaces of vacua: more explicit examples

In this appendix, we give some specific examples of our main results for $\mathrm{SQCD}[N_c, k, l, n_f, 0]$. In particular, we display the intricate matching of the vacua across the maximally- and minimally-chiral dualities.

## B.1  Crossing $\xi = 0$

Here are a few examples of phase transitions from $\xi > 0$ to $\xi < 0$, for various values of the parameters $[N_c, k, l, n_f]$:

| $N_c$ | $k$ | $l$ | $n_f$ | $\xi > 0$ phase | $\xi < 0$ phase |
|---|---|---|---|---|---|
| 3 | $\frac{5}{2}$ | $-5$ | 7 | $\mathrm{Gr}(3,7) \oplus U(3)_{6,-9}$ | $\underbrace{U(2)_{6,-4} \times U(1)_{-6}}_{-5}$ |
| 4 | 6 | $-3$ | 8 | $\mathrm{Gr}(4,8) \oplus U(4)_{10,-2}$ | $\mathrm{Gr}(3,8) \times U(1)_{-1} \oplus \mathrm{Gr}(2,8) \times U(2)_{2,-4}$ |
| 5 | $\frac{11}{2}$ | 0 | 9 | $\mathrm{Gr}(5,9) \oplus \mathrm{Gr}(4,9) \times U(1)_1$ | $U(5)_{10,10}$ |
| 6 | 4 | $-2$ | 10 | $\mathrm{Gr}(6,10) \oplus U(6)_{9,-3}$ | $\underbrace{U(5)_{9,-1} \times U(1)_{-3}}_{-2}$ |
| 7 | $\frac{3}{2}$ | -5 | 11 | $\mathrm{Gr}(7,11) \oplus \underbrace{U(5)_{7,-18} \times U(2)_{-4,-14}}_{-5}$ $\oplus\, U(7)_{7,-28} \oplus \underbrace{U(6)_{7,-23} \times U(1)_{-9}}_{-5}$ | $\underbrace{U(3)_{7,-8} \times U(4)_{-4,-24}}_{-5}$ $\oplus\, \underbrace{U(4)_{7,-13} \times U(3)_{-4,-19}}_{-5}$ |
| 8 | 5 | 6 | 10 | $\mathrm{Gr}(8,10) \oplus \mathrm{Gr}(7,10) \times U(1)_7$ | $U(8)_{11,59}$ |
| 9 | 8 | $-7$ | 14 | $\mathrm{Gr}(9,14) \oplus U(9)_{15,-48}$ | $\mathrm{Gr}(8,14) \times U(1)_{-6}$ |
| 10 | 9 | -4 | 12 | $\mathrm{Gr}(10,12) \oplus U(10)_{15,-25}$ | $\mathrm{Gr}(9,12) \times U(1)_{-1} \oplus \mathrm{Gr}(8,12) \times U(2)_{3,-5}$ $\oplus\, \mathrm{Gr}(7,12) \times U(3)_{3,-9}$ |
| 11 | $-\frac{9}{2}$ | -4 | 13 | $\mathrm{Gr}(11,13)$ | $U(11)_{-11,-55} \oplus \underbrace{U(1)_{-2} \times U(10)_{-11,-51}}_{-4}$ $\oplus\, \underbrace{U(2)_{2,-6} \times U(9)_{-11,-47}}_{-4}$ |
| 12 | $\frac{15}{3}$ | 21 | 21 | $\mathrm{Gr}(12,21)$ | $U(12)_{18,18} \oplus \underbrace{U(9)_{18,18} \times U(3)_{-3,-3}}_{0}$ $\oplus \underbrace{U(10)_{18,18} \times U(2)_{-3,-3}}_{0}$ $\oplus \underbrace{U(11)_{18,18} \times U(1)_{-3}}_{0}$ |
| 30 | 1 | 0 | 32 | $\mathrm{Gr}(30,32)$ | $\underbrace{U(15)_{17,17} \times U(15)_{-15,-15}}_{0}$ $\oplus \underbrace{U(16)_{17,17} \times U(14)_{-15,-15}}_{0}$ $\oplus \underbrace{U(17)_{-17,-17} \times U(13)_{-15,-15}}_{0}$ |

## B.2 Minimally-chiral duality for $\xi > 0$

Here we consider a few examples of the minimally-chiral duality with positive FI parameter:

| $N_c$ | $k$ | $l$ | $n_f$ | Electric Vacua | Magnetic Vacua |
|---|---|---|---|---|---|
| 3 | $\frac{9}{2}$ | -5 | 7 | $\mathrm{Gr}(3,7)$ <br><br> $\oplus\, U(3)_{8,-7}$ | $\mathrm{Gr}(4,7)\times \underbrace{U(1)_0 \times U(1)_{-4}}_{1}$ <br><br> $\oplus \underbrace{U(5)_{-8,-3}\times U(1)_{-4}}_{1}$ |
| 4 | 7 | -2 | 8 | $\mathrm{Gr}(4,8)$ <br><br> $\oplus\, \mathrm{Gr}(3,8)\times U(1)_1$ | $\mathrm{Gr}(4,8)\times \underbrace{U(3)_{-3,0}\times U(1)_{-1}}_{1}$ <br><br> $\oplus\, \mathrm{Gr}(5,8)\times \underbrace{U(2)_{-3,-1}\times U(1)_{-1}}_{1}$ |
| 5 | $-\frac{13}{2}$ | 5 | 9 | $\mathrm{Gr}(5,9)$ <br><br> $\oplus\, U(5)_{-11,14}$ | $\mathrm{Gr}(4,9)\times \underbrace{U(2)_{2,0}\times U(1)_4}_{-1}$ <br><br> $\oplus \underbrace{U(6)_{11,5}\times U(1)_4}_{-1}$ |
| 6 | 6 | -5 | 8 | $\mathrm{Gr}(6,8)$ <br><br> $\oplus\, U(6)_{10,-20}$ | $\mathrm{Gr}(2,8)\times \underbrace{U(2)_{-2,0}\times U(1)_{-4}}_{1}$ <br><br> $\oplus \underbrace{U(4)_{-10,-6}\times U(1)_{-4}}_{1}$ |
| 11 | 12 | 0 | 12 | $\mathrm{Gr}(11,12)$ <br><br> $\oplus\, \mathrm{Gr}(10,12)\times U(1)_6$ <br><br> $\oplus\, \mathrm{Gr}(9,12)\times U(2)_{6,6}$ <br><br> $\oplus\, \mathrm{Gr}(8,12)\times U(3)_{6,6}$ <br><br> $\oplus\, \mathrm{Gr}(7,12)\times U(4)_{6,6}$ <br><br> $\oplus\, \mathrm{Gr}(6,12)\times U(5)_{6,6}$ <br><br> $\oplus\, \mathrm{Gr}(5,12)\times U(6)_{6,6}$ | $\mathbb{CP}^{11}\times \underbrace{U(6)_{-6,0}\times U(1)_1}_{1}$ <br><br> $\oplus\, \mathrm{Gr}(2,12)\times \underbrace{U(5)_{-6,-1}\times U(1)_1}_{1}$ <br><br> $\oplus\, \mathrm{Gr}(3,12)\times \underbrace{U(4)_{-6,-2}\times U(1)_1}_{1}$ <br><br> $\oplus\, \mathrm{Gr}(4,12)\times \underbrace{U(3)_{-6,-3}\times U(1)_1}_{1}$ <br><br> $\oplus\, \mathrm{Gr}(5,12)\times \underbrace{U(2)_{-6,-4}\times U(1)_1}_{1}$ <br><br> $\oplus\, \mathrm{Gr}(6,12)\times \underbrace{U(1)_{-5}\times U(1)_1}_{1}$ <br><br> $\oplus\, \mathrm{Gr}(7,12)\times U(1)_1$ |
| 12 | $\frac{23}{2}$ | -10 | 21 | $\mathrm{Gr}(12,21)$ <br><br> $\oplus\, U(12)_{22,-98}$ | $\mathrm{Gr}(9,21)\times \underbrace{U(1)_0 \times U(1)_{-9}}_{1}$ <br><br> $\oplus \underbrace{U(10)_{-22,-12}\times U(1)_{-9}}_{1}$ |
| 20 | $\frac{33}{2}$ | 0 | 27 | $\mathrm{Gr}(20,27)$ <br><br> $\oplus\, \mathrm{Gr}(19,27)\times U(1)_3$ <br><br> $\oplus\, \mathrm{Gr}(18,27)\times U(2)_{3,3}$ <br><br> $\oplus\, \mathrm{Gr}(17,27)\times U(3)_{3,3}$ | $\mathrm{Gr}(7,27)\times \underbrace{U(3)_{-3,0}\times U(1)_1}_{1}$ <br><br> $\oplus\, \mathrm{Gr}(8,27)\times \underbrace{U(2)_{-3,-1}\times U(1)_1}_{1}$ <br><br> $\oplus\, \mathrm{Gr}(9,27)\times \underbrace{U(1)_{-2}\times U(1)_1}_{1}$ <br><br> $\oplus\, \mathrm{Gr}(10,27)\times U(1)_1$ |

## B.3 Maximally-chiral duality for $\xi > 0$

Here consider we consider a few examples of the maximally-chiral duality with positive FI parameter:

| $N_c$ | $k$ | $l$ | $n_f$ | Electric Vacua | Magnetic Vacua |
|---|---|---|---|---|---|
| 3 | $\frac{3}{2}$ | $-5$ | 7 | $\mathrm{Gr}(3,7) \oplus U(3)_{5,-10}$ | $\mathrm{Gr}(4,7) \oplus \underbrace{U(2)_{-5,-15} \times U(2)_{2,-8}}_{-5}$ |
| 4 | 3 | $-7$ | 8 | $\mathrm{Gr}(4,8) \oplus U(4)_{7,-21}$ | $\mathrm{Gr}(4,8) \oplus \underbrace{U(3)_{-7,-28} \times U(1)_{-6}}_{-7}$ |
| 5 | $-3$ | $-5$ | 10 | $\mathrm{Gr}(5,10) \oplus \underbrace{U(2)_{2,-8} \times U(3)_{-8,-23}}_{-5}$ | $\mathrm{Gr}(5,10) \oplus U(5)_{8,-17}$ |
| 6 | 3 | -12 | 10 | $\mathrm{Gr}(6,10)$ $\oplus U(6)_{8,-64}$ $\oplus \underbrace{U(5)_{8,-52} \times U(1)_{-14}}_{-12}$ | $\mathrm{Gr}(4,10)$ $\oplus \underbrace{U(2)_{-8,-32} \times U(2)_{2,-22}}_{-12}$ $\oplus \underbrace{U(3)_{-8,-44} \times U(1)_{-10}}_{-12}$ |
| 14 | -7 | 4 | 20 | $\mathrm{Gr}(14,20)$ $\oplus U(14)_{-17,39}$ $\oplus \underbrace{U(1)_7 \times U(13)_{-17,35}}_{4}$ | $\mathrm{Gr}(6,20)$ $\oplus \underbrace{U(3)_{-3,9} \times U(3)_{17,29}}_{4}$ $\oplus \underbrace{U(2)_{-3,5} \times U(4)_{17,33}}_{4}$ |
| 15 | $\frac{11}{2}$ | 4 | 23 | $\mathrm{Gr}(15,23) \oplus \underbrace{U(9)_{17,53} \times U(6)_{-6,18}}_{4}$ | $\mathrm{Gr}(8,23) \oplus U(8)_{-17,15}$ |
| 17 | 0 | 10 | 20 | $\mathrm{Gr}(17,20) \oplus \underbrace{U(7)_{10,80} \times U(10)_{-10,90}}_{10}$ | $\mathrm{Gr}(3,20) \oplus U(3)_{-10,20}$ |
| 20 | $\frac{13}{2}$ | -4 | 25 | $\mathrm{Gr}(20,25)$ $\oplus \underbrace{U(17)_{19,-49} \times U(3)_{-6,-18}}_{-4}$ $\oplus \underbrace{U(18)_{19,-53} \times U(2)_{-6,-14}}_{-4}$ $\oplus \underbrace{U(19)_{19,-57} \times U(1)_{-10}}_{-4}$ | $\mathrm{Gr}(5,25)$ $\oplus \underbrace{U(2)_{-19,-27} \times U(3)_{6,-6}}_{-4}$ $\oplus \underbrace{U(1)_{-23} \times U(4)_{6,-10}}_{-4}$ $\oplus U(5)_{6,-14}$ |
| 23 | 19 | -23 | 42 | $\mathrm{Gr}(23,42)$ $\oplus U(23)_{40,-489}$ $\oplus \underbrace{U(22)_{40,-466} \times U(1)_{-25}}_{-23}$ | $\mathrm{Gr}(19,42)$ $\oplus \underbrace{U(17)_{-40,-431} \times U(2)_{2,-44}}_{-23}$ $\oplus \underbrace{U(18)_{-40,-454} \times U(1)_{-21}}_{-23}$ |
| 30 | $\frac{11}{2}$ | 2 | 45 | $\mathrm{Gr}(30,45) \oplus \underbrace{U(13)_{28,54} \times U(17)_{-17,17}}_{2}$ | $\mathrm{Gr}(15,45) \oplus U(15)_{-28,2}$ |
| 40 | 9 | -3 | 46 | $\mathrm{Gr}(40,46)$ $\oplus \underbrace{U(32)_{32,-64} \times U(8)_{-14,-38}}_{-3}$ | $\mathrm{Gr}(6,46)$ $\oplus U(6)_{14,-4}$ |

## B.4 Minimally-chiral duality for $\xi < 0$

Next, we consider we consider a few examples of the minimally-chiral duality with negative FI parameter:

| $N_c$ | $k$ | $l$ | $n_f$ | Electric Vacua | Magnetic Vacua |
|---|---|---|---|---|---|
| 3 | $\frac{9}{2}$ | $-5$ | 7 | $\mathrm{Gr}(2,7)\times U(1)_{-4}$ | $\mathrm{Gr}(5,7)\times U(1)_{-4}$ |
| 4 | 7 | -2 | 8 | $\mathrm{Gr}(2,8)\times U(2)_{3,-1}$ $\oplus \mathbb{CP}^7\times U(3)_{3,-3}$ $\oplus U(4)_{11,3}$ | $\mathrm{Gr}(6,8)\times \underbrace{U(1)_{-2}\times U(1)_{-1}}_{1}$ $\oplus \mathrm{Gr}(7,8)\times U(1)_{-1}$ $\oplus \underbrace{U(7)_{-11,-4}\times U(1)_{-1}}_{1}$ |
| 5 | $-\frac{13}{2}$ | 5 | 9 | $\mathrm{Gr}(4,9)\times U(1)_3$ $\oplus \mathrm{Gr}(3,9)\times U(2)_{-2,8}$ | $\oplus \mathrm{Gr}(5,9)\times \underbrace{U(1)_1\times U(1)_4}_{-1}$ $\mathrm{Gr}(6,9)\times U(1)_4$ |
| 6 | 6 | -5 | 8 | $\mathrm{Gr}(5,8)\times U(1)_{-3}$ $\oplus \mathrm{Gr}(4,8)\times U(2)_{2,-8}$ | $\mathrm{Gr}(3,8)\times \underbrace{U(1)_{-1}\times U(1)_{-4}}_{1}$ $\oplus \mathrm{Gr}(4,8)\times U(1)_{-4}$ |
| 11 | 12 | 0 | 12 | $U(11)_{18,18}$ | $\underbrace{U(7)_{-18,-11}\times U(1)_1}_{1}$ |
| 12 | $\frac{23}{2}$ | $-10$ | 21 | $\mathrm{Gr}(11,21)\times U(1)_{-9}$ | $\mathrm{Gr}(10,21)\times U(1)_{-9}$ |
| 14 | 14 | -7 | 24 | $\mathrm{Gr}(13,24)\times U(1)_{-5}$ $\oplus \mathrm{Gr}(12,24)\times U(2)_{2,-12}$ | $\mathrm{Gr}(11,24)\times \underbrace{U(1)_{-1}\times U(1)_{-6}}_{1}$ $\oplus \mathrm{Gr}(12,24)\times U(1)_{-6}$ |
| 23 | $\frac{29}{2}$ | -10 | 25 | $\mathrm{Gr}(22,25)\times U(1)_{-8}$ $\oplus \mathrm{Gr}(21,25)\times U(2)_{2,-18}$ | $\mathrm{Gr}(3,25)\times \underbrace{U(1)_{-1}\times U(1)_{-9}}_{1}$ $\oplus \mathrm{Gr}(4,25)\times U(1)_{-9}$ |
| 37 | $\frac{53}{2}$ | -5 | 45 | $\mathrm{Gr}(36,45)\times U(1)_{-1}$ $\oplus \mathrm{Gr}(35,45)\times U(2)_{4,-6}$ $\oplus \mathrm{Gr}(34,45)\times U(3)_{4,-11}$ $\oplus \mathrm{Gr}(33,45)\times U(4)_{4,-16}$ | $\mathrm{Gr}(9,45)\times \underbrace{U(3)_{-4,-1}\times U(1)_{-4}}_{1}$ $\oplus \mathrm{Gr}(10,45)\times \underbrace{U(2)_{-4,-2}\times U(1)_{-4}}_{1}$ $\oplus \mathrm{Gr}(11,45)\times \underbrace{U(1)_{-3}\times U(1)_{-4}}_{1}$ $\oplus \mathrm{Gr}(12,45)\times U(1)_{-4}$ |
| 40 | -28 | 11 | 50 | $\mathrm{Gr}(39,50)\times U(1)_8$ $\oplus \mathrm{Gr}(38,50)\times U(2)_{-3,19}$ $\oplus \mathrm{Gr}(37,50)\times U(3)_{-3,30}$ | $\mathrm{Gr}(11,50)\times \underbrace{U(2)_{3,1}\times U(1)_{10}}_{-1}$ $\oplus \mathrm{Gr}(12,50)\times \underbrace{U(1)_2\times U(1)_{10}}_{-1}$ $\oplus \mathrm{Gr}(13,50)\times U(1)_{10}$ |
| 52 | $\frac{77}{2}$ | -3 | 73 | $\mathrm{Gr}(51,73)\times U(1)_{-1}$ $\oplus \mathrm{Gr}(50,73)\times U(2)_{2,-4}$ | $\mathrm{Gr}(22,73)\times \underbrace{U(1)_{-1}\times U(1)_{-2}}_{1}$ $\oplus \mathrm{Gr}(23,73)\times U(1)_{-2}$ |

### B.5 Maximally-chiral duality for $\xi < 0$

Finally, here are a few examples of the maximally-chiral duality with negative FI parameter:

| $N_c$ | $k$ | $l$ | $n_f$ | Electric Vacua | Magnetic Vacua |
|---|---|---|---|---|---|
| 3 | $\frac{3}{2}$ | −5 | 7 | $U(1)_0 \times \underbrace{U(2)_{-2,-12}}_{-5} \oplus \underbrace{U(2)_{5,-5} \times U(1)_{-7}}_{-5}$ | $U(4)_{-5,-25} \oplus \underbrace{U(3)_{-5,-20} \times U(1)_{-3}}_{-5}$ |
| 4 | 3 | −7 | 8 | $\underbrace{U(3)_{7,-14} \times U(1)_{-8}}_{-7}$ | $U(4)_{-7,-35}$ |
| 5 | -3 | -5 | 10 | $U(5)_{-8,-33}$ $\oplus \underbrace{U(1)_{-3} \times U(4)_{-8,-28}}_{-5}$ | $\underbrace{U(2)_{-2,-12} \times U(3)_{8,-7}}_{-5}$ $\oplus \underbrace{U(1)_{-7} \times U(4)_{8,-12}}_{-5}$ |
| 6 | 3 | −12 | 10 | $\underbrace{U(4)_{8,-40} \times U(2)_{-2,-26}}_{-12}$ | $U(4)_{-8,-56}$ |
| 14 | -7 | 4 | 20 | $\underbrace{U(2)_{3,11} \times U(12)_{-17,31}}_{4}$ $\oplus \underbrace{U(3)_{3,15} \times U(11)_{-17,27}}_{4}$ | $\underbrace{U(1)_1 \times U(5)_{17,37}}_{4}$ $\oplus U(6)_{17,41}$ |
| 15 | $\frac{11}{2}$ | 4 | 23 | $U(15)_{17,77}$ $\oplus \underbrace{U(10)_{17,57} \times U(5)_{-6,14}}_{4}$ $\oplus \underbrace{U(11)_{17,61} \times U(4)_{-6,10}}_{4}$ $\oplus \underbrace{U(12)_{17,65} \times U(3)_{-6,6}}_{4}$ $\oplus \underbrace{U(13)_{17,69} \times U(2)_{-6,2}}_{4}$ $\oplus \underbrace{U(14)_{17,73} \times U(1)_{-2}}_{4}$ | $\underbrace{U(2)_{-17,-9} \times U(6)_{6,30}}_{4}$ $\oplus \underbrace{U(7)_{-17,11} \times U(1)_{10}}_{4}$ $\oplus \underbrace{U(6)_{-17,7} \times U(2)_{6,14}}_{4}$ $\oplus \underbrace{U(5)_{-17,3} \times U(3)_{6,18}}_{4}$ $\oplus \underbrace{U(4)_{-17,-1} \times U(4)_{6,22}}_{4}$ $\oplus \underbrace{U(3)_{-17,-5} \times U(5)_{6,26}}_{4}$ |
| 23 | 19 | −23 | 42 | $\underbrace{U(21)_{40,-443} \times U(2)_{-2,-48}}_{-23}$ | $U(19)_{-40,-477}$ |
| 29 | 14 | 0 | 30 | $U(29)_{29,29} \oplus U(28)_{29,29} \times U(1)_{-1}$ | $U(1)_1 \oplus U(1)_{-29}$ |
| 35 | 20 | -3 | 50 | $\underbrace{U(30)_{45,-45} \times U(5)_{-5,-20}}_{-3}$ $\oplus \underbrace{U(31) \times U(4)_{-5,-17}}_{-3}$ $\oplus \underbrace{U(32)_{45,-51} \times U(3)_{-5,-14}}_{-3}$ | $U(15)_{-45,-90}$ $\oplus \underbrace{U(14)_{-45,-87} \times U(1)_2}_{-3}$ $\oplus \underbrace{U(13)_{-45,-84} \times U(2)_{5,-1}}_{-3}$ |
| 43 | -20 | 10 | 52 | $\underbrace{U(5)_{6,56} \times U(38)_{-46,334}}_{10}$ $\oplus \underbrace{U(6)_{6,66} \times U(37)_{-46,324}}_{10}$ | $\underbrace{U(1)_4 \times U(8)_{46,126}}_{10}$ $\oplus U(9)_{46,136}$ |

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
