# Peer review of "On the Witten index of 3d $\mathcal{N}=2$ unitary SQCD with general CS levels"

_SciPost Physics, doi:SciPost Phys. 15, 085 (2023)_

## Round 2 · Referee Report · Anonymous (Referee 1) · 2023-6-22

Strengths

1) Provides a novel, explicit and extensive computation of a fundamental observable (Witten index) of an important and well-studied class of 3d N=2 theories (unitary SQCD)
2) Presents an interesting exploration of the dependence of the vacuum moduli spaces on real deformation parameters/phase transitions, on which the computation is based
3) Contains a satisfactory presentation of consistency checks of the results (matching previously known results and dualities)
4) Proposes a discussion of a seemingly new generalisation of a known dualities for pure Chern-Simons theories
5) Excellent language and formatting
6) It includes a very welcome Mathemtica package implementing the computations

Weaknesses

1) The computation is based on geometric assumptions that would perhaps deserve a more extensive explanation 2) The conjectural kind of strongly-coupled vacua remain obscure 3) The paper contains no conclusion or discussion of future directions

Report

This is an interesting paper that offers an extensive computation of the Witten index of 3d N=2 SQCD with an arbitrary number of fundamental and anti-fundamental chiral multiplets and values of Chern-Simons levels in the presence of non-vanishing FI parameters.

The computation is based on 1) an iterative argument that relates the index of such a theory to the index of a theory with no anti-fundamental multiplets and 2) a discussion of the vacuum geometry in the latter case, especially with respect to topological and Higgs vacua. 1) is based on making some specific mass very large (therefore integrating out some of the multiplets). The discussion of 2) is quite detailed, but some of the assumptions could have been clarified better.

At each value of FI parameter possible contributions of Higgs and topological vacua are discussed. At critical levels it is conjectured that some strongly-coupled vacua must exist in order to ensure the invariance of the index under deformations. However, a direct check of their contribution remains out of reach.

The proposed closed formulas pass several consistency checks including 1) agreement with previous formulas of the same authors derived from a Bethe analysis when these can be computed and 2) consistency with some recently proposed dualities. A companion Mathematica notebook presents these calculations. The new dualities for pure Chern-Simons theory are also interesting.

The vacuum geometry of N=2 theories is somewhat underdeveloped and this paper constitutes a welcome contribution, with results that pass solid consistency checks.

I would like to request some changes in the form of 1) clarifications and 2) typos. I will also make some optional suggestions.

Requested changes

Clarifications:

p.4, bottom: please clarify that the invariance of the Witten index is confirmed for non-critical Chern-Simons levels only, and that it is otherwise used to conjecture the existence of extra vacua
p.4, below 1.4: SU(N) or PSU(N)?
p.9 To make the paper self-contained, please briefly mention what is meant by 3d A-model in this context (as for example done in the author’s previous paper). This is also relevant for the invariance of the Witten index
p.11, first paragraph: same as for p.4, should the flavour symmetry not contain PSU(N) factors instead of SU(N)?
p.12, around equation 3.10: Is there a priori reason why the fibration must be trivial? If so, please provide the reason. If not, please state that 3.10 is a working assumption, and explain whether it could impact the conjectural existence of strongly-coupled vacua. Maybe introduce here the notation used at p.17 (Gr(…)xU(…))
p.16, below 4.3: Since the Witten index is graded by Fermion number, it would be useful to state explicitly why geometric vacua are indeed a lower bound (i.e. why all other contributions are positive)
p.17 Further on the point related to p.12, the different types of vacua may be connected to each other (e.g. sigma can be continuously tuned to certain special values where the type changes), but the computations seem to treat all the types as independent/disjoint (sum type II and III). Is it clear that we can simply sum all contributions?
p.17 Please briefly explain why SU(p), and mention the invariance of the Witten index that is used to conjecture the existence of these vacua
p.18, 4.7: the second inequality does not seem to be explicitly motivated at this point (there is a related discussion at p.23)
p.27, 5.2 (notational, pedantic): epsilon is previously defined as a sign, please say \pm 1

There are a few typos in the paper, including:

p.11 beginning of point ii): consist
p.14 eq.34: The Q’s should have i and j indices
p.16, last paragraph: “What we call Type II vacua”  What we call Type II,a vacua
p.23, first paragraph: “Comes from requiring that the volume of the Grassmannian manifold has a positive”
p.23, last paragraph: “to be non-zeros”

Suggestions:

1) The definition of the Witten index is given in words and an equation is not provided until p.10. I would suggest to bring that equation forward
2) A paper published in this journal, https://scipost.org/10.21468/SciPostPhys.3.3.024, provides a detailed description of the geometry of the moduli space of N=4 SQCD . Since the set-up is closely related, it could be worth mentioning it and pointing out similarities
3) The absence of conclusions is excused since the paper is well-written and the importance of the Witten index clear. I would however recommend including some possible future directions
4) The equations governing the existence of conjectural strongly-coupled vacua at a critical level is reminiscent of the wall-crossing formulas for the topologically twisted index proposes by the follwing cited paper, which also appeared in this journal: https://scipost.org/10.21468/SciPostPhys.12.6.186 . It could be interesting to relate these phenomena.
5) The Mathematica Notebook, which is very welcome, would benefit from the inclusion of some initialisation cells as well as some formatting (but it’s excellent as it is)

  • validity: high
  • significance: good
  • originality: good
  • clarity: high
  • formatting: perfect
  • grammar: perfect

Author:  Cyril Closset  on 2023-07-10  [id 3788]

(in reply to Report 1 on 2023-06-22)

We thank the referee for the detailed comments and suggestions. Here are short replies to the two most important points:

  1. The global form of the flavour group is somewhat subtle. In the case of n_a=0, the "Grassmannian theory", the symmetry is SU(n_f), as can be seen explicitly by computing the twisted index (wherein all irreps of SU(n_f) appear). Interestingly, the irreps that appear are correlated with the flux sector (so with the topological charge). This needs further study, and so this is something we hope to explore further in future work.

  2. about comments about p17: at fixed masses m and/or FI parameter, the vacua are not continuously connected to each other, they are simply the solutions to the vacuum equations. The sigma is not a tunable parameter, since it is the parameter we solve for. Hence we should indeed just sum over all solutions/vacua.

---

## Round 2 · Referee Report · Anonymous (Referee 2) · 2023-6-26

Report

This paper studied the moduli space of vacua of 3d $\mathcal{N}=2$ SQCD using Witten index. In particular, a novel recursion relation of the Witten index is derived, enabling the author to derive an explicit formula for the Witten index, which is quite unusual.

The paper is well written and clear so I recommend this paper for publication.

Requested changes

It would be nice to add some clarifying words recalling the relation and difference between Witten index, moduli space of vacua/vacuum structure and Bethe vacua.

Typo: on page 16, "to be pedagogical, we will first analysise the"

  • validity: -
  • significance: -
  • originality: -
  • clarity: -
  • formatting: -
  • grammar: -

Author:  Cyril Closset  on 2023-07-10  [id 3789]

(in reply to Report 2 on 2023-06-26)

We thank the referee for their comments and suggestions.

---

## Round 2 · Referee Report · Anonymous (Referee 3) · 2023-6-29

Report

The paper seems to be interesting, but I did not follow recent developments in this field and it is difficult for me to judge on its scientific validity. Especially because the paper is written not in a user-friendly self-contained way.
For example, the authors do not write the Lagrangian of what they call U(N)_{k,k+lN} model and the notation that they use is not clear. Well, I've looked into the references [5] and [27] and understood what they meant, but the explanations should be given in the text of the paper.
A question to the authors: they consider only N=2 theories, but can they apply their methods alsl for N=1 theories and reproduce and/or generalize the results of Ref. [11] ?
I also bring to their attention a review 1312.1804, where both pure N=1 CS-SYM theories and also the theories including matter were discussed.
  • validity: -
  • significance: -
  • originality: -
  • clarity: -
  • formatting: -
  • grammar: -

Author:  Cyril Closset  on 2023-07-10  [id 3790]

(in reply to Report 3 on 2023-06-29)

We thank the referee for their comments and suggestions. We will add some small clarifications regarding the definition of the CS levels.

We did not study the N=1 case, however there is some overlap with some previous work as cited in the text. (see comment at the middle of page 4.) Most of our methods do not work for N=1 because we need a Coulomb branch. Moreover, the underlying reason for the mass-deformation-invariance of the N=2 Witten index is the holomorphy of the vacuum moduli space when compatified in 2d, as explained by Intriligator and Seiberg in 2013; this is not available with only 2 supercharges in 3d.

---

## Round 3 · List of Changes

We corrected various typos, including all the ones pointed out by the referees. We also added various small clarifications to address the suggestions and concerned raised in the reports. Here is a list of the most salient changes:

1) We added equation (1.2) and clarified the discussion around it.

2) We added a comment about future directions at the end of the intro.

3) We added footnote 4 about the flavour group.

4) We clarified the notation above eq.(3.9)

5) We clarified why we have some (S)U(p)_0 groups appearing on page 18.

---

## Editorial Decision

published